# Simultaneous Swap Regret Minimization via KL-Calibration

**Haipeng Luo**[*]
USC
haipengl@usc.edu

**Spandan Senapati**[*]
USC
ssenapat@usc.edu

**Vatsal Sharan**[*]
USC
vsharan@usc.edu

## Abstract

Calibration is a fundamental concept that aims at ensuring the reliability of probabilistic predictions by aligning them with real-world outcomes. There is a surge of studies on new calibration measures that are easier to optimize compared to the classical $\ell_1$-Calibration while still having strong implications for downstream applications. One such recent example is the work by Fishelson et al. (2025) who show that it is possible to achieve $\tilde{\mathcal{O}}(T^{1/3})$ pseudo $\ell_2$-Calibration error via minimizing pseudo swap regret of the squared loss, which in fact implies the same bound for all bounded proper losses with a smooth univariate form. In this work, we significantly generalize their result in the following ways: (a) in addition to smooth univariate forms, our algorithm also simultaneously achieves $\tilde{\mathcal{O}}(T^{1/3})$ swap regret for any proper loss with a twice continuously differentiable univariate form (such as Tsallis entropy); (b) our bounds hold not only for pseudo swap regret that measures losses using the forecaster's distributions on predictions, but also hold for the actual swap regret that measures losses using the forecaster's actual realized predictions.

We achieve so by introducing a new stronger notion of calibration called *(pseudo) KL-Calibration*, which we show is equivalent to the (pseudo) swap regret with respect to log loss. We prove that there exists an algorithm that achieves $\tilde{\mathcal{O}}(T^{1/3})$ KL-Calibration error and provide an explicit algorithm that achieves $\tilde{\mathcal{O}}(T^{1/3})$ pseudo KL-Calibration error. Moreover, we show that the same algorithm achieves $\mathcal{O}(T^{1/3}(\log T)^{-\frac{1}{3}}\log(T/\delta))$ swap regret with probability at least $1 - \delta$ for any proper loss with a smooth univariate form, which implies $\tilde{\mathcal{O}}(T^{1/3})$ $\ell_2$-Calibration error. A technical contribution of our work is a new randomized rounding procedure and a non-uniform discretization scheme to minimize the swap regret for log loss.

## 1 Introduction

We consider online *calibration* — a problem of making sequential probabilistic predictions over binary outcomes. Formally, at each time $t = 1, \ldots, T$, a forecaster randomly predicts $p_t \in [0, 1]$ while simultaneously the adversary chooses $y_t \in \{0, 1\}$, and subsequently the forecaster observes the true label $y_t$. Letting $n_p$ denote the number of rounds the forecaster predicts $p_t = p$, the forecaster's predictions are perfectly calibrated if for all $p \in [0, 1]$, the empirical distribution of the label conditioned on the forecast being $p$, i.e., the quantity $\rho_p := \sum_{t:p_t=p} y_t/n_p$, matches $p$. The $\ell_q$-Calibration error ($q \geq 1$) is then defined as $\mathsf{Cal}_q := \sum_{p \in [0,1]} \sum_{t=1}^{T} \mathbb{I}[p_t = p] (p - \rho_p)^q$.

A related concept used in Fishelson et al. (2025) that we call *pseudo calibration error* measures the error using the forecaster's conditional distribution $\mathcal{P}_t \in \Delta_{[0,1]}$ at time $t$, instead of the actual prediction $p_t$. More specifically, the pseudo $\ell_q$-Calibration error is defined as $\mathsf{PCal}_q := \sum_{t=1}^{T} \mathbb{E}_{p \sim \mathcal{P}_t}[(p - \tilde{\rho}_p)^q]$,

---

[*]Author ordering is alphabetical.

39th Conference on Neural Information Processing Systems (NeurIPS 2025).

where $\tilde{\rho}_p := \frac{\sum_{t=1}^{T} y_t \mathcal{P}_t(p)}{\sum_{t=1}^{T} \mathcal{P}_t(p)}$. By not dealing with the random variable $p_t$, pseudo calibration is often easier to optimize.

Two of the most popular calibration measures are $\ell_1$ and $\ell_2$-Calibration. It has been long known that $\mathsf{Cal}_1 = \mathcal{O}(T^{2/3})$ is achievable, and there are some recent breakthroughs towards closing the gap between this upper bound and a standard lower bound $\mathsf{Cal}_1 = \Omega(\sqrt{T})$ (see more discussion in related work). For $\ell_2$-Calibration, Foster and Hart (2023) proposed an algorithm based on the concept of "calibeating" that achieves $\mathbb{E}[\mathsf{Cal}_2] = \tilde{\mathcal{O}}(T^{\frac{1}{3}})$. Moreover, a recent work by Fishelson et al. (2025) showed that $\mathsf{PCal}_2 = \tilde{\mathcal{O}}(T^{\frac{1}{3}})$ is achievable by establishing equivalence to pseudo swap regret of the squared loss and proposing an efficient algorithm based on the well-known Blum-Mansour reduction (Blum and Mansour, 2007) for minimizing pseudo swap regret.

More specifically, given a loss function $\ell : [0,1] \times \{0,1\} \to \mathbb{R}$, the swap regret of the forecaster is defined as $\mathsf{SReg}^\ell := \sup_{\sigma:[0,1]\to[0,1]} \mathsf{SReg}^\ell_\sigma$, where $\mathsf{SReg}^\ell_\sigma := \sum_{t=1}^{T} \ell(p_t, y_t) - \ell(\sigma(p_t), y_t)$ measures the difference between the forecaster's total loss and the loss of a strategy that always swaps the forecaster's prediction via a swap function $\sigma$. Similarly, pseudo swap regret (Fishelson et al., 2025; referred in their work as full swap regret) is defined using the conditional distribution of predictions $\mathcal{P}_t$ instead of $p_t$ itself: $\mathsf{PSReg}^\ell := \sup_{\sigma:[0,1]\to[0,1]} \mathsf{PSReg}^\ell_\sigma$, where $\mathsf{PSReg}^\ell_\sigma := \sum_{t=1}^{T} \mathbb{E}_{p\sim\mathcal{P}_t}[\ell(p, y_t) - \ell(\sigma(p), y_t)]$. Fishelson et al. (2025) show that it is possible to achieve $\mathsf{PSReg}^\ell = \tilde{\mathcal{O}}(T^{\frac{1}{3}})$ when $\ell$ is the squared loss, which, as we will show, further implies that the same bound holds for any bounded proper loss $\ell$ with a smooth univariate form (refer to Section 2 for concrete definitions of proper losses and their univariate form).

In this work, we significantly generalize their results by not only recovering their results for pseudo swap regret, but also proving the same $\tilde{\mathcal{O}}(T^{\frac{1}{3}})$ bound for new losses such as log loss and those induced by the Tsallis entropy. Moreover, we prove the same bound (either in expectation or with high probability) for the actual swap regret, which was missing in Fishelson et al. (2025). To achieve these goals, we introduce a natural notion of *(pseudo) KL-Calibration*, where the penalty incurred by the forecaster's prediction $p$ deviating from the empirical distribution of $y$ (conditioned on the forecast being $p$) is measured in terms of the KL-divergence. Specifically, the KL-Calibration and the pseudo KL-Calibration incurred by the forecaster are respectively defined as

$$\mathsf{KLCal} := \sum_{p\in[0,1]} \sum_{t=1}^{T} \mathbb{I}[p_t = p]\mathsf{KL}(\rho_p, p), \quad \mathsf{PKLCal} := \sum_{t=1}^{T} \mathbb{E}_{p\sim\mathcal{P}_t}[\mathsf{KL}(\tilde{\rho}_p, p)], \qquad (1)$$

where $\mathsf{KL}(q,p) = q\log\frac{q}{p} + (1-q)\log\frac{1-q}{1-p}$ is the KL-divergence for two Bernoulli distributions with mean $q$ and $p$ respectively. It follows from Pinsker's inequality that $\mathsf{KL}(\rho_p, p) \geq (\rho_p - p)^2$, therefore, $\mathsf{KLCal} \geq \mathsf{Cal}_2$ and $\mathsf{PKLCal} \geq \mathsf{PCal}_2$, making (pseudo) KL-Calibration a stronger measure for studying upper bounds than (pseudo) $\ell_2$-Calibration.

**Contributions and Technical Overview** Let $\mathcal{L}$ denote the class of bounded (in $[-1,1]$) proper losses. Our concrete contributions are as follows.

- In Section 3, we start by discussing the implications of (pseudo) KL-Calibration towards minimizing (pseudo) swap regret. In particular, in subsection 3.1, we show for each $\ell \in \mathcal{L}_2$, where $\mathcal{L}_2$ is the class of bounded proper losses whose univariate form $\ell(p) := \mathbb{E}_{y\sim p}[\ell(p,y)]$ is twice continuously differentiable in $(0,1)$, we have $\mathsf{SReg}^\ell = \mathcal{O}(\mathsf{KLCal}), \mathsf{PSReg}^\ell = \mathcal{O}(\mathsf{PKLCal})$. In subsection 3.2, we show that for each $\ell \in \mathcal{L}_G$, where $\mathcal{L}_G$ is the class of bounded proper losses with a $G$-smooth univariate form, (pseudo) KL-Calibration implies that $\mathsf{SReg}^\ell \leq G \cdot \mathsf{Cal}_2 \leq G \cdot \mathsf{KLCal}, \mathsf{PSReg}^\ell \leq G \cdot \mathsf{PCal}_2 \leq G \cdot \mathsf{PKLCal}$. This gives us strong incentives to study $\mathsf{PKLCal}$ and $\mathsf{KLCal}$.

- In Section 4, we prove that there exists an algorithm that achieves $\mathbb{E}[\mathsf{KLCal}] = \mathcal{O}(T^{\frac{1}{3}}(\log T)^{\frac{5}{3}})$. To achieve so, we first realize that (pseudo) KL-Calibration is equivalent to the (pseudo) swap regret of the log loss $\ell(p,y) = -y\log p - (1-y)\log(1-p)$, i.e., $\mathsf{KLCal} = \mathsf{SReg}^\ell, \mathsf{PKLCal} = \mathsf{PSReg}^\ell$. Subsequently, we propose a non-constructive proof for minimizing $\mathsf{SReg}^\ell$; our proof is based on swapping the forecaster and the adversary via von-Neumann's minimax theorem. Two particularly technical aspects of our proof are the usage of a non uniform discretization, which is contrary to all previous works, and the use of Freedman's inequality for martingale difference sequences.

We remark that our non-constructive proof is motivated from Hu and Wu (2024), who provide a similar proof to show the existence of an algorithm that simultaneously achieves $\mathcal{O}(\sqrt{T}\log T)$ swap regret for any bounded proper loss. However, compared to Hu and Wu (2024), we use a non uniform discretization, which requires a more involved analysis.[*] Moreover, due to the desired $\mathcal{O}(T^{\frac{1}{3}})$ nature of our final bounds, we cannot merely use Azuma-Hoeffding that guarantees $\mathcal{O}(\sqrt{T})$ concentration. The aforementioned reasons combined make our analysis considerably non-trivial and different than Hu and Wu (2024).

Combined with the implications of Section 3, we show the existence of an algorithm that simultaneously achieves the following bounds on $\mathbb{E}[\mathsf{SReg}^\ell]$: (a) $\mathcal{O}(T^{\frac{1}{3}}(\log T)^{\frac{5}{3}})$ for the log loss; (b) $\mathcal{O}(T^{\frac{1}{3}}(\log T)^{\frac{5}{3}})$ for each $\ell \in \mathcal{L}_2$; (c) $\mathcal{O}(G \cdot T^{\frac{1}{3}}(\log T)^{\frac{5}{3}})$ for each $\ell \in \mathcal{L}_G$; and (d) $\mathcal{O}(T^{\frac{2}{3}}(\log T)^{\frac{5}{6}})$ for each $\ell \in \mathcal{L}\backslash\{\mathcal{L}_2 \cup \mathcal{L}_G\}$. Notably, our result is better than Luo et al. (2024) who studied the weaker notion of external regret, defined as $\mathrm{REG}^\ell := \sup_{p \in [0,1]} \sum_{t=1}^T \ell(p_t, y_t) - \ell(p, y_t)$, and showed that the Follow-the-Leader (FTL) algorithm achieves $\mathrm{REG}^\ell = \mathcal{O}(\log T)$ for each $\ell \in \mathcal{L}_2 \cup \mathcal{L}_G$, however incurs $\mathrm{REG}^\ell = \Omega(T)$ for a specific $\ell \in \mathcal{L}\backslash\{\mathcal{L}_2 \cup \mathcal{L}_G\}$.

- In Section 5, we propose an explicit algorithm that achieves $\mathsf{PKLCal} = \mathcal{O}(T^{\frac{1}{3}}(\log T)^{\frac{2}{3}})$. Similar to Fishelson et al. (2025), we utilize the Blum-Mansour reduction for minimizing $\mathsf{PSReg}^\ell$ for the log loss. However, our key novelty lies in the usage of a non uniform discretization and a new randomizing rounding procedure (Algorithm 4) for the log loss. Since the log loss is not Lipschitz, we show that the common rounding schemes studied in the literature fail to work for our considered discretization. A natural implication of our result is that, since $\mathsf{PSReg}^\ell \leq G \cdot \mathsf{PKLCal}$ for any $\ell \in \mathcal{L}_G$, we recover the result of Fishelson et al. (2025). However, since $\mathsf{PSReg}^\ell = \mathcal{O}(\mathsf{PKLCal})$ for any $\ell \in \mathcal{L}_2$, we are able to deal with new losses, and even the log loss which is unbounded.

- Finally, in Appendix E, we show that if we only consider the class of bounded proper losses with a smooth univariate form, our algorithm guarantees

$$\mathsf{Cal}_2 = \mathcal{O}\left(T^{1/3}(\log T)^{-\frac{1}{3}}\log(T/\delta)\right), \quad \mathsf{Msr}_{\mathcal{L}_G} = \mathcal{O}\left(G \cdot T^{1/3}(\log T)^{-\frac{1}{3}}\log(T/\delta)\right)$$

with probability at least $1 - \delta$, where $\mathsf{Msr}_{\mathcal{L}_G} = \sup_{\ell \in \mathcal{L}_G} \mathsf{SReg}^\ell$. This marks the first appearance of a sub-$\sqrt{T}$ high probability bound for classical $\ell_2$-Calibration via an efficient algorithm.

**Related Work**  Calibration can also be viewed from the lens of simultaneous regret minimization (Kleinberg et al., 2023; Hu and Wu, 2024; Luo et al., 2024). It is known from Kleinberg et al. (2023) that $\ell_1$-Calibrated forecasts can simultaneously lead to sublinear swap regret for all $\ell \in \mathcal{L}$, where recall that $\mathcal{L}$ is the class of bounded (in $[-1, 1]$) proper losses. However, as shown by Qiao and Valiant (2021); Dagan et al. (2024), for any forecasting algorithm there exists an adversary that ensures that $\mathsf{Cal}_1 = \Omega(T^{0.54389})$, thereby sidestepping the goal of achieving the favorable $\sqrt{T}$ style regret guarantee. Despite the limitations of calibration, Hu and Wu (2024) proposed an explicit algorithm that achieves $\mathbb{E}[\sup_{\ell \in \mathcal{L}} \mathsf{SReg}^\ell] = \mathcal{O}(\sqrt{T}\log T)$. Compared to (Hu and Wu, 2024), we show that a single algorithm in fact achieves $\tilde{\mathcal{O}}(T^{\frac{1}{3}})$ swap regret for important subclasses of $\mathcal{L}$ and even the log loss, while simultaneously achieving $\tilde{\mathcal{O}}(T^{\frac{2}{3}})$ swap regret for any arbitrary $\ell \in \mathcal{L}$. Notably, the result of Hu and Wu (2024) does not apply to the log loss since it does not belong to $\mathcal{L}$. With an appropriate post-processing of the predictions, a stronger analogue of simultaneous swap regret minimization has also been studied in the contextual setting (Garg et al., 2024; referred to as swap omniprediction), where the forecaster competes with functions from a hypothesis class $\mathcal{F}$. Notably, in swap omniprediction, both the loss function and the competing hypothesis are parameterized by the predictions themselves. For this, Garg et al. (2024) showed that it is impossible to achieve $\mathcal{O}(\sqrt{T})$ swap omniprediction error for the class of convex and Lipschitz loss functions, even in the simplest setting where $\mathcal{F}$ contains the constant $0, 1$ functions. Additional related work is deferred to Appendix A.

---

[*] Our non-uniform discretization scheme has appeared before (Kotłowski et al., 2016), albeit in a different context. Its combination with other techniques in our paper results in a significantly different approach.

## 2 Preliminaries and Background

**Notation** For a $m \in \mathbb{N}$, $[m]$ denotes the index set $\{1, \ldots, m\}$. We reserve bold lower-case alphabets for vectors and bold upper-case alphabets for matrices. The notation $\mathbb{I}[\cdot]$ refers to the indicator function, which evaluates to $1$ if the condition is true, and $0$ otherwise. We use $\boldsymbol{e}_i$ to represent the $i$-th standard basis vector (dimension inferred from context), which is $1$ at the $i$-th coordinate and $0$ everywhere else. For any $k \in \mathbb{N}$, we use $\Delta_k$ to represent the $(k-1)$-dimensional simplex. Moreover, we use $\Delta_{[0,1]}$ to represent the set of all probability distributions over $[0,1]$. We use $\mathbb{P}_t, \mathbb{E}_t$ to represent the conditional probability, expectation respectively, where the conditioning is over the randomness till time $t-1$ (inclusive). We use $\mathsf{KL}(p,q), \mathsf{TV}(p,q), \chi^2(p,q)$ to represent the KL divergence, total variation distance, chi-squared distance between two Bernoulli distributions with means $p, q$. For a set $\mathcal{I}$, its complement is $\bar{\mathcal{I}} = \Omega \backslash \mathcal{I}$, where the sample set $\Omega$ shall be clear from the context. A twice differentiable function $f : \mathcal{D} \to \mathbb{R}$ is $\alpha$-*smooth* over $\mathcal{D} \subset \mathbb{R}$ if $f''(x) \leq \alpha$ for all $x \in \mathcal{D}$. A function $f : \mathcal{W} \to \mathbb{R}$ is $\alpha$-*exp-concave* over a convex set $\mathcal{W}$ if the function $\exp(-\alpha f(w))$ is concave over $\mathcal{W}$. We use the notation $\tilde{\mathcal{O}}(\cdot)$ to hide lower order logarithmic terms.

**Proper Losses** A loss $\ell : [0,1] \times \{0,1\} \to \mathbb{R}$ is called proper if $\mathbb{E}_{y \sim p}[\ell(p, y)] \leq \mathbb{E}_{y \sim p}[\ell(p', y)]$ for all $p, p' \in [0,1]$. Intuitively, a proper loss incentivizes the forecaster to report the true distribution of the label. Throughout the paper, we shall be primarily concerned about the family $\mathcal{L}$ (or a subset) of bounded proper losses, i.e., $\mathcal{L} := \{\ell \text{ s.t. } \ell \text{ is proper and } \ell(p,y) \in [-1,1] \text{ for all } p \in [0,1], y \in \{0,1\}\}$, even though our results hold for (and in fact achieved via) the unbounded log loss. For a proper loss $\ell$, the *univariate* form of $\ell$ is defined as $\ell(p) := \mathbb{E}_{y \sim p}[\ell(p, y)]$. It turns out that a the univariate form of a proper loss is concave. Moreover, one can construct a proper loss using a concave univariate form based on the following characterization lemma.

**Lemma 1** (Theorem 2 in Gneiting and Raftery (2007)). *A loss $\ell : [0,1] \times \{0,1\} \to \mathbb{R}$ is proper if and only if there exists a concave function $f$ such that $\ell(p, y) = f(p) + \langle g_p, y - p \rangle$ for all $p \in [0,1], y \in \{0,1\}$, where $g_p$ denotes a subgradient of $f$ at $p$. Also, $f$ is the univariate form of $\ell$.*

Examples of proper losses include squared loss $\ell(p, y) = (p - y)^2$, log loss $\ell(p, y) = y \log \frac{1}{p} + (1 - y) \log \frac{1}{1-p}$, spherical loss $\ell(p, y) = -\frac{py + (1-p)(1-y)}{\sqrt{p^2 + (1-p)^2}}$, etc.

**Bregman Divergence** For a convex function $\phi$, let $\mathsf{BREG}_\phi(x, y) = \phi(x) - \phi(y) - \langle \partial\phi(y), x - y \rangle$ denote the Bregman divergence associated with $\phi$. The following lemma is important to our results.

**Lemma 2** (Lemma 3.8 in Hu and Wu (2024)). *Let $u : [0,1] \to [-1,1]$ be a twice differentiable concave function. Then, we have $\mathsf{BREG}_{-u}(\hat{p}, p) = \int_p^{\hat{p}} |u''(\mu)| \cdot (\hat{p} - \mu) d\mu$.*

**Problem Setting** As mentioned in Section 1, we consider calibration, where the interaction between the forecaster and the adversary is according to the following protocol: at each time $t = 1, \ldots, T$, (a) the forecaster randomly predicts $p_t \in [0,1]$ and simultaneously the adversary chooses $y_t \in \{0,1\}$; (b) the forecaster observes $y_t$. Throughout the paper, we shall consider algorithms that make predictions $p_t$ that fall in a finite discretization $\mathcal{Z} \subset [0,1]$. According to (1), the KL-Calibration, Pseudo KL-Calibration incurred by the forecaster are $\mathsf{KLCal} = \sum_{p \in \mathcal{Z}} \sum_{t=1}^{T} \mathbb{I}[p_t = p] \mathsf{KL}(\rho_p, p)$, $\mathsf{PKLCal} = \sum_{p \in \mathcal{Z}} \sum_{t=1}^{T} \mathcal{P}_t(p) \mathsf{KL}(\tilde{\rho}_p, p)$, where $\rho_p = \frac{\sum_{t=1}^{T} y_t \mathbb{I}[p_t = p]}{\sum_{t=1}^{T} \mathbb{I}[p_t = p]}, \tilde{\rho}_p = \frac{\sum_{t=1}^{T} y_t \mathcal{P}_t(p)}{\sum_{t=1}^{T} \mathcal{P}_t(p)}$.[*] For simplicity, we assume that the adversary is oblivious, that is it selects $y_1, \ldots, y_T$ at time $t = 0$ with complete knowledge of the forecaster's algorithm.[*] Our goal is to minimize the (pseudo) KL-Calibration error, which as we show in Section 3, has powerful implications.

As mentioned, the swap regret of the forecaster with respect to a loss function $\ell$ against a swap function $\sigma : [0,1] \to [0,1]$ is $\mathsf{SReg}_\sigma^\ell = \sum_{t=1}^{T} \ell(p_t, y_t) - \ell(\sigma(p_t), y_t)$. Swap regret is then defined as $\mathsf{SReg}^\ell = \sup_{\sigma:[0,1] \to [0,1]} \mathsf{SReg}_\sigma^\ell$. Similarly, the pseudo swap regret is $\mathsf{PSReg}^\ell = \sup_{\sigma:[0,1] \to [0,1]} \mathsf{PSReg}_\sigma^\ell$, where $\mathsf{PSReg}_\sigma^\ell = \sum_{p \in \mathcal{Z}} \sum_{t=1}^{T} \mathcal{P}_t(p)(\ell(p, y_t) - \ell(\sigma(p), y_t))$. We further define *maximum (pseudo) swap regret* with respect to the class of bounded proper losses $\mathcal{L}$ as

---

[*] For convenience, we set $\frac{0}{0} = 0$. This is because if $n_p = 0$, the forecast $p_t = p$ was never made and thus does not contribute to the calibration error.

[*] However, our results generalize directly to an adaptive adversary who decides $y_t$ based on $p_1, \ldots, p_{t-1}$.

$\mathsf{Msr}_{\mathcal{L}} := \sup_{\ell \in \mathcal{L}} \mathsf{SReg}^{\ell}, \mathsf{PMsr}_{\mathcal{L}} := \sup_{\ell \in \mathcal{L}} \mathsf{PSReg}^{\ell}$. For a subset of losses $\mathcal{L}' \subseteq \mathcal{L}$, we define $\mathsf{Msr}_{\mathcal{L}'}$ and $\mathsf{PMsr}_{\mathcal{L}'}$ similarly, with the supremum over $\ell \in \mathcal{L}'$. The usage of $\ell$ for a bounded proper loss, or the log loss (which does not belong to $\mathcal{L}$) shall be clear from the context.

# 3 Implications of (Pseudo) KL-Calibration

In this section, we discuss the implications of (pseudo) KL-Calibration towards minimizing the (pseudo) swap regret. In particular, we shall show that (pseudo) KL-Calibration upper bounds the following: (a) (P)$\mathsf{SReg}^{\ell}$ for all $\ell \in \mathcal{L}_2$ (subsection 3.1); (b) (P)$\mathsf{Msr}_{\mathcal{L}_G}$ (subsection 3.2). This gives a strong incentive to study (pseudo) KL-Calibration.

The following proposition, which relates (pseudo) swap regret with Bregman Divergence is central to all subsequent results developed in this work.

**Proposition 1.** *For any proper loss $\ell$ and a swap function $\sigma : [0,1] \to [0,1]$, let $\mathsf{BREG}_{-\ell}$ be the Bregman divergence associated with the negative univariate form $-\ell$. We have*

$$\mathsf{SReg}_{\sigma}^{\ell} = \sum_{p \in \mathcal{Z}} \left( \sum_{t=1}^{T} \mathbb{I}[p_t = p] \right) \left( \mathsf{BREG}_{-\ell}(\rho_p, p) - \mathsf{BREG}_{-\ell}(\rho_p, \sigma(p)) \right),$$

$$\mathsf{PSReg}_{\sigma}^{\ell} = \sum_{p \in \mathcal{Z}} \left( \sum_{t=1}^{T} \mathcal{P}_t(p) \right) \left( \mathsf{BREG}_{-\ell}(\tilde{\rho}_p, p) - \mathsf{BREG}_{-\ell}(\tilde{\rho}_p, \sigma(p)) \right),$$

*where $\rho_p = \frac{\sum_{t=1}^{T} \mathbb{I}[p_t = p] y_t}{\sum_{t=1}^{T} \mathbb{I}[p_t = p]}, \tilde{\rho}_p = \frac{\sum_{t=1}^{T} \mathcal{P}_t(p) y_t}{\sum_{t=1}^{T} \mathcal{P}_t(p)}$. Furthermore,*

$$\mathsf{SReg}^{\ell} = \sum_{p \in \mathcal{Z}} \sum_{t=1}^{T} \mathbb{I}[p_t = p] \mathsf{BREG}_{-\ell}(\rho_p, p), \quad \mathsf{PSReg}^{\ell} = \sum_{p \in \mathcal{Z}} \sum_{t=1}^{T} \mathcal{P}_t(p) \mathsf{BREG}_{-\ell}(\tilde{\rho}_p, p).$$

The proof of Proposition 1, deferred to Appendix B, follows by an application of Lemma 1 and is similar to Hu and Wu (2024). Two particularly interesting applications of Proposition 1 are:

- For the squared loss $\ell(p, y) = (p-y)^2$, the univariate form is $\ell(p) = p - p^2$, and $\mathsf{BREG}_{-\ell}(\rho_p, p) = (\rho_p - p)^2$. Therefore, $\mathsf{SReg}^{\ell} = \mathsf{Cal}_2, \mathsf{PSReg}^{\ell} = \mathsf{PCal}_2$.

- For the log loss $\ell(p, y) = y \log \frac{1}{p} + (1-y) \log \frac{1}{1-p}$, the univariate form is $\ell(p) = \mathbb{E}_{y \sim p}[\ell(p, y)] = -p \log p - (1-p) \log(1-p)$. Moreover, as can be verified by direct computation, the associated Bregman divergence $\mathsf{BREG}_{-\ell}(\hat{p}, p)$ is exactly equal to $\mathsf{KL}(\hat{p}, p)$. Therefore, we have $\mathsf{SReg}^{\ell} = \mathsf{KLCal}, \mathsf{PSReg}^{\ell} = \mathsf{PKLCal}$. This equivalence between (pseudo) KL-Calibration and (pseudo) swap regret of the log loss shall be our starting tool towards the developments in Sections 4, 5, where we bound $\mathsf{KLCal}, \mathsf{PKLCal}$ respectively.

Note that since $\mathsf{PSReg}^{\ell} \leq \mathbb{E}[\mathsf{SReg}^{\ell}]$ trivially holds by definition, $\mathsf{PCal}_2$ and $\mathsf{PKLCal}$ are indeed weaker notions compared to $\mathsf{Cal}_2$ and $\mathsf{KLCal}$ respectively.

## 3.1 (Pseudo) KL-Calibration implies (pseudo) swap regret for all $\ell \in \mathcal{L}_2$

In this subsection, we show that $\mathsf{SReg}^{\ell} = \mathcal{O}(\mathsf{KLCal}), \mathsf{PSReg}^{\ell} = \mathcal{O}(\mathsf{PKLCal})$ for each $\ell \in \mathcal{L}_2$, where

$$\mathcal{L}_2 := \{\ell \in \mathcal{L} \text{ s.t. the univariate form } \ell(p) \text{ is twice continuously differentiable in } (0,1)\}.$$

Note that according to Lemma 1, for all $\ell \in \mathcal{L}$, the univariate form must be concave, Lipschitz, and bounded, for the induced loss $\ell(p, y)$ to be proper and bounded. In addition to these implicit constraints, we require the condition that the second derivative $\ell''(p)$ is continuous in $(0,1)$. We state several examples of losses that belong to $\mathcal{L}_2$. First, the squared loss clearly belongs to $\mathcal{L}_2$, since its univariate form is $\ell(p) = p - p^2$. Second, consider a generalization of the squared loss via Tsallis entropy, which corresponds to a loss with the univariate form $\ell(p) = -c \cdot p^{\alpha}$, where we choose $\alpha > 1$ and the proportionality constant $c > 0$ is to ensure that the induced loss $\ell(p, y)$ is in $[-1, 1]$ (refer Lemma 1). We have, $\ell(p, y) = c(\alpha - 1)p^{\alpha} - \alpha c p^{\alpha-1} y$, which is in $\mathcal{L}_2$. Third, the spherical loss has the univariate form $\ell(p) = -\sqrt{p^2 + (1-p)^2}$ and is also contained in $\mathcal{L}_2$.

The following lemma, derived by Luo et al. (2024), provides a growth rate on the second derivative of any $\ell \in \mathcal{L}_2$ and is a key ingredient for our proof of the desired implication.

**Lemma 3** (Lemma 2 in Luo et al. (2024)). *For a function $f$ that is concave, Lipschitz, and bounded over $[0,1]$ and twice continuously differentiable over $(0,1)$, there exists a constant $c > 0$ such that $|f''(p)| \le c \cdot \max\left(\frac{1}{p}, \frac{1}{1-p}\right)$ for all $p \in (0,1)$.*

Using this to bound $|u''(p)|$ in the statement of Lemma 2, we immediately obtain the following proposition whose proof can be found in Appendix B.

**Proposition 2.** *Let $\ell \in \mathcal{L}_2$. Then, we have $\mathsf{BREG}_{-\ell}(\hat{p}, p) = \mathcal{O}\left(\mathsf{KL}(\hat{p}, p)\right)$ and thus*

$$\mathsf{SReg}^\ell = \mathcal{O}(\mathsf{KLCal}), \quad \mathsf{PSReg}^\ell = \mathcal{O}(\mathsf{PKLCal}).$$

Note the constant $c_\ell$ hidden in the $\mathcal{O}(.)$ notation in the result above is exactly the constant guaranteed by Lemma 3, which is finite. However, this is not sufficient to conclude that $\sup_{\ell \in \mathcal{L}_2} c_\ell < \infty$ (since $\mathcal{L}_2$ is infinite), therefore, we do not necessarily guarantee that $(\mathsf{P})\mathsf{Msr}_{\mathcal{L}_2}$ (defined as $\sup_{\ell \in \mathcal{L}_2}(\mathsf{P})\mathsf{SReg}^\ell$) is $\mathcal{O}((\mathsf{P})\mathsf{KLCal})$. We remark that this is only a minor technical issue (that has also implicitly appeared in the prior work of Luo et al. (2024)), and our result in Proposition 2 implies that (pseudo) KL-Calibration simultaneously bounds (pseudo) swap regret for all $\ell \in \mathcal{L}_2$. This in itself is quite meaningful and perfectly aligns with the goal in downstream decision making — to guarantee diminishing (swap) regret for all loss functions simultaneously. Henceforth, all subsequent results related to (pseudo) swap regret for $\mathcal{L}_2$ are stated similarly. We also remark that Proposition 2 holds more generally for any subclass of proper losses where each loss satisfies the growth rate in Lemma 3. To keep the exposition simple, we only state our results for $\mathcal{L}_2$.

### 3.2 (Pseudo) KL-Calibration implies (pseudo) maximum swap regret against $\mathcal{L}_G$

We now consider another class $\mathcal{L}_G$, containing proper losses whose univariate form is $G$-smooth, i.e., $\mathcal{L}_G := \{\ell \in \mathcal{L} \text{ s.t. } |\ell''(p)| \le G \text{ for all } p \in [0,1]\}$. Losses that belong to $\mathcal{L}_G$ include squared loss, spherical loss, Tsallis entropy for $\alpha \ge 2$, etc. Notably, the latter does not lie in $\mathcal{L}_G$ for $\alpha \in (1,2)$. Using Lemma 2 again, along with the fact $\mathsf{PCal}_2 \le \mathsf{PKLCal}, \mathsf{Cal}_2 \le \mathsf{KLCal}$ due to Pinsker's inequality, we immediately obtain the following.

**Proposition 3.** *Let $\ell \in \mathcal{L}_G$. Then, we have $\mathsf{BREG}_{-\ell}(\hat{p}, p) \le G(\hat{p} - p)^2$, and thus*

$$\mathsf{Msr}_{\mathcal{L}_G} \le G \cdot \mathsf{Cal}_2 \le G \cdot \mathsf{KLCal}, \quad \mathsf{PMsr}_{\mathcal{L}_G} \le G \cdot \mathsf{PCal}_2 \le G \cdot \mathsf{PKLCal}.$$

The proof of Proposition 3 is deferred to Appendix B. As already mentioned, Fishelson et al. (2025) proposed an algorithm that achieves $\mathsf{PCal}_2 = \tilde{\mathcal{O}}(T^{\frac{1}{3}})$, which implies that the same algorithm in fact ensures $\mathsf{PMsr}_{\mathcal{L}_G} = \tilde{\mathcal{O}}(G \cdot T^{\frac{1}{3}})$. However, the implications of $\mathsf{KLCal}, \mathsf{PKLCal}$ allow us get simultaneous guarantees for a broader subclass of proper losses, particularly, $\mathcal{L}_2 \cup \mathcal{L}_G$.

## 4 Achieving KL-Calibration

In this section, we prove that there exists an algorithm that achieves $\mathbb{E}[\mathsf{SReg}^\ell] = \mathcal{O}(T^{\frac{1}{3}}(\log T)^{\frac{5}{3}})$ for $\ell$ being the log loss, therefore the same algorithm achieves $\mathbb{E}[\mathsf{KLCal}] = \mathcal{O}(T^{\frac{1}{3}}(\log T)^{\frac{5}{3}})$. Our proof is non-constructive, since it is based on swapping the adversary and the algorithm via the minimax theorem (Theorem 3 in Appendix C), and deriving a forecasting algorithm in the dual game.

**Theorem 1.** *There exists an algorithm that achieves $\mathbb{E}[\mathsf{SReg}^\ell] = \mathcal{O}(T^{\frac{1}{3}}(\log T)^{\frac{5}{3}})$ for the log loss, where the expectation is taken over the internal randomness of the algorithm.*

The proof of Theorem 1 is quite technical and is deferred to Appendix C. We discuss the key novelty of our proof here. Two particularly technical aspects of our proof are the usage of a non uniform discretization, which is contrary to all previous works, and the use of Freedman's inequality for martingale difference sequences (Lemma 8). In particular, we employ the following discretization scheme: $\mathcal{Z} = \{z_1, \dots, z_{K-1}\} \subset [0,1]$, where $z_i = \sin^2\left(\frac{\pi i}{2K}\right)$ and $K \in \mathbb{N}$ is a constant to be specified later. For convinience, we set $z_0 = 0, z_K = 1$, however, $z_0, z_K$ are not included in the discretization. For our analysis, we require a discretization scheme that satisfies the following

constraints: (a) $z_i - z_{i-1} = \mathcal{O}(\frac{1}{K})$ for all $i \in [K]$; (b) $\frac{\max^2(z_i - z_{i-1}, z_{i+1} - z_i)}{z_i(1-z_i)} = \mathcal{O}\left(\frac{1}{K^2}\right)$ for all $i \in [K-1]$; (c) $\sum_{i=1}^{K-1} \frac{1}{z_i(1-z_i)} = \mathcal{O}(K^2)$; and (d) $\sum_{i=1}^{K-1} \frac{1}{\sqrt{z_i(1-z_i)}} = \tilde{\mathcal{O}}(K)$. The uniform discretization $\mathcal{Z} = \{\frac{1}{K}, \ldots, \frac{K-1}{K}\}$ satisfies (a), (c), (d) above, however, doesn't satisfy (b). As we show in Lemma 6 (Appendix C), our considered non uniform discretization achieves all these required bounds by having a finer granularity close to the boundary of $[0, 1]$, thereby making it suitable for our purpose. The following steps provide a brief sketch of our proof, which is proved for an adaptive adversary and therefore also holds for the weaker oblivious adversary.

**Step I**   We only consider discretized forecasters that make predictions that lie inside $\mathcal{Z}$. Since the strategy space of such forecasters is finite, and that of the adversary is trivially finite, the minimax theorem (Theorem 3) applies, and we can swap the adversary and the algorithm, thereby resulting in the dual game. In this dual game, at every time $t$, the adversary first reveals the conditional distribution of $y_t$, based on which the forecaster predicts $p_t$. We consider a forecaster $F$ which at time $t$ does the following: (a) it computes $\tilde{p}_t = \mathbb{E}_t[y_t]$; (b) predicts $p_t = \mathrm{argmin}_{z \in \mathcal{Z}} |\tilde{p}_t - z|$. For such a forecaster, we obtain a high probability bound on $\mathsf{SReg}^\ell$, and subsequently bound $\mathbb{E}[\mathsf{SReg}^\ell]$.

**Step II**   Applying Lemma 8, we show that for each $i$ (with $n_i = n_{z_i}$)

$$\left| \sum_{t=1}^{T} \mathbb{I}[p_t = z_i](\tilde{p}_t - y_t) \right| \leq 2\sqrt{\log \frac{2}{\delta}} \cdot \max\left( \sqrt{n_i \left( z_i(1-z_i) + \frac{\pi}{2K} \right)}, \sqrt{\log \frac{2}{\delta}} \right)$$

with probability at least $1 - \delta KT$. Using this, we bound $|z_i - \rho_i|$, where $\rho_i$ is a shorthand for $\rho_{z_i}$. Notably, the bound above dictates separate consideration of $i \in \mathcal{I}$ and $i \in \bar{\mathcal{I}}$ (depending on which term realizes the maximum), where $\mathcal{I} := \left\{ i \in [K-1]; n_i < \frac{\log \frac{2}{\delta}}{z_i(1-z_i) + \frac{\pi}{2K}} \right\}$.

**Step III**   Next, we write $\mathsf{SReg}^\ell$ as the sum of two terms $\mathsf{SReg}^\ell = \text{Term I} + \text{Term II}$, where Term I $= \sum_{i \in \mathcal{I}} n_i \mathsf{KL}(\rho_i, z_i)$, Term II $= \sum_{i \in \bar{\mathcal{I}}} n_i \mathsf{KL}(\rho_i, z_i)$, and bound Term I, II individually. Since $\mathsf{KL}(\rho_i, z_i) \leq \chi^2(\rho_i, z_i) = \frac{(\rho_i - z_i)^2}{z_i(1-z_i)}$, we utilize the bound on $|\rho_i - z_i|$ obtained in the previous step and show that Term II $= \mathcal{O}\left( \frac{T}{K^2} + K \log \frac{1}{\delta} \right)$. Importantly, the use of Freedman's inequality provides a variance term that mitigates the potentially small denominator of $\frac{(\rho_i - z_i)^2}{z_i(1-z_i)}$. Similarly, we show that Term I $= \mathcal{O}\left( \frac{T}{K^2} + K(\log K)^{\frac{3}{2}} \log \frac{1}{\delta} \right)$. Combining, we obtain $\mathsf{SReg}^\ell = \mathcal{O}\left( \frac{T}{K^2} + K(\log K)^{\frac{3}{2}} \log \frac{1}{\delta} \right)$ with probability at least $1 - \delta KT$. Subsequently, we bound $\mathbb{E}[\mathsf{SReg}^\ell]$ by setting $\delta = 1/T, K = T^{\frac{1}{3}}/(\log T)^{\frac{5}{6}}$.

Equipped with Theorem 1, we prove the following stronger corollary (proof deferred to Appendix C).

**Corollary 1.** *There exists an algorithm that achieves the following bounds simultaneously:*

$$\mathbb{E}\left[\mathsf{KLCal}\right] = \mathcal{O}(T^{\frac{1}{3}}(\log T)^{\frac{5}{3}}), \quad \mathbb{E}\left[\mathsf{Msr}_{\mathcal{L}_G}\right] = \mathcal{O}(G \cdot T^{\frac{1}{3}}(\log T)^{\frac{5}{3}}),$$

$$\mathbb{E}\left[\mathsf{Msr}_{\mathcal{L}_2}\right] = \mathcal{O}(T^{\frac{1}{3}}(\log T)^{\frac{5}{3}}), \quad \mathbb{E}\left[\mathsf{Msr}_{\mathcal{L}\backslash\{\mathcal{L}_G \cup \mathcal{L}_2\}}\right] = \mathcal{O}(T^{\frac{2}{3}}(\log T)^{\frac{5}{6}}),$$

*where the expectation is taken over the internal randomness of the algorithm.*

# 5   Achieving Pseudo KL-Calibration

In this section, we propose an explicit algorithm that achieves $\mathsf{PSReg}^\ell = \mathcal{O}(T^{\frac{1}{3}}(\log T)^{\frac{2}{3}})$ for the log loss, therefore the same algorithm achieves $\mathsf{PKLCal} = \mathcal{O}(T^{\frac{1}{3}}(\log T)^{\frac{2}{3}})$. Our algorithm is based on the well-known Blum-Mansour (BM) reduction (Blum and Mansour, 2007) and extends the idea from Fishelson et al. (2025). First, we employ a similar but slightly different non uniform discretization scheme that adds two extra end points $z_0$ and $z_K$ to the one used in the previous section (for technical reasons):

$$\mathcal{Z} = \{z_0, z_1, \ldots, z_{K-1}, z_K\}, \text{where } z_0 = \sin^2 \frac{\pi}{4K}, z_i = \sin^2 \frac{\pi i}{2K} \text{ for } i \in [K-1], z_K = \cos^2 \frac{\pi}{4K},$$

and $K \in \mathbb{N}$ is a constant to be specified later. The same scheme was used before by Rooij and Erven (2009); Kotłowski et al. (2016) for different problems. Since the conditional distribution $\mathcal{P}_t$ has support over $\mathcal{Z}$, taking supremum over all swap functions $\sigma : \mathcal{Z} \to \mathcal{Z}$ in Proposition 1, we obtain

$$\sup_{\sigma:\mathcal{Z}\to\mathcal{Z}} \mathrm{PSReg}^\ell_\sigma = \mathrm{PSReg}^\ell - \sum_{p\in\mathcal{Z}}\sum_{t=1}^{T}\mathcal{P}_t(p)\inf_{\sigma:\mathcal{Z}\to\mathcal{Z}}\mathrm{BREG}_{-\ell}(\rho_p,\sigma(p)) \geq \mathrm{PSReg}^\ell - \frac{(2-\sqrt{2})\pi^2 T}{K^2},$$

where the inequality follows by choosing $\sigma(p) = \mathrm{argmin}_{z\in\mathcal{Z}}\mathrm{BREG}_{-\ell}(\rho_p,z)$. For this choice of $\sigma$, from Kotłowski et al. (2016, page 13), we have $\mathrm{BREG}_{-\ell}(\rho_p,\sigma(p)) \leq (2-\sqrt{2})\frac{\pi^2}{K^2}$. Therefore,

$$\mathrm{PSReg}^\ell \leq \sup_{\sigma:\mathcal{Z}\to\mathcal{Z}}\mathrm{PSReg}^\ell_\sigma + (2-\sqrt{2})\pi^2\frac{T}{K^2}, \tag{2}$$

and it suffices to bound $\sup_{\sigma:\mathcal{Z}\to\mathcal{Z}}\mathrm{PSReg}^\ell_\sigma$, which we do via the BM reduction. Towards this end, we first recall the BM reduction. The reduction maintains $K+1$ external regret algorithms $\mathcal{A}_0,\ldots,\mathcal{A}_K$. At each time $t$, let $\boldsymbol{q}_{t,i} \in \Delta_{K+1}$ represent the probability distribution over $\mathcal{Z}$ output by $\mathcal{A}_i$. Let $\boldsymbol{Q}_t = [\boldsymbol{q}_{t,0},\ldots,\boldsymbol{q}_{t,K}]$ be the matrix obtained by stacking the vectors $\boldsymbol{q}_{t,0},\ldots,\boldsymbol{q}_{t,K}$ as columns. We compute the stationary distribution of $\boldsymbol{Q}_t$, i.e., a distribution $\boldsymbol{p}_t \in \Delta_{K+1}$ over $\mathcal{Z}$ that satisfies $\boldsymbol{Q}_t\boldsymbol{p}_t = \boldsymbol{p}_t$. With $\boldsymbol{p}_t$ being our final distribution of predictions (that is, $\mathcal{P}_t(z_i) = p_{t,i}$), we draw a prediction from it and observe $y_t$. After that, we feed the scaled loss function $p_{t,i}\ell(.,y_t)$ to $\mathcal{A}_i$. Let $\tilde{\boldsymbol{\ell}}_{t,i} = p_{t,i}\boldsymbol{\ell}_t \in \mathbb{R}^{K+1}$ be a scaled loss vector, where $\ell_t(j) = \ell(z_j,y_t)$. It then follows from Blum and Mansour (2007, Theorem 5) that $\sup_{\sigma:\mathcal{Z}\to\mathcal{Z}}\mathrm{PSReg}^\ell_\sigma \leq \sum_{i=0}^{K}\mathrm{REG}_i$, where $\mathrm{REG}_i :=$ $\sup_{j\in[K+1]}\sum_{t=1}^{T}\langle\boldsymbol{q}_{t,i}-\boldsymbol{e}_j,\tilde{\boldsymbol{\ell}}_{t,i}\rangle$, i.e., the pseudo swap regret is bounded by the sum of the external regrets of the $K+1$ algorithms. We summarize the discussion so far in Algorithm 1.

---

**Algorithm 1** BM for log loss

**Initialize:** $\mathcal{A}_i$ for $i \in \{0,\ldots,K\}$ and set $\boldsymbol{q}_1 = \left[\frac{1}{K+1},\ldots,\frac{1}{K+1}\right]$;

1: **for** $t = 1,\ldots,T$
2:     Set $\boldsymbol{Q}_t = [\boldsymbol{q}_{t,0},\ldots,\boldsymbol{q}_{t,K}]$;
3:     Compute the stationary distribution of $\boldsymbol{Q}_t$, i.e., $\boldsymbol{p}_t \in \Delta_{K+1}$ that satisfies $\boldsymbol{Q}_t\boldsymbol{p}_t = \boldsymbol{p}_t$;
4:     Output conditional distribution $\mathcal{P}_t$, where $\mathcal{P}_t(z_i) = p_t(i)$ and observe $y_t$;
5:     **for** $i = 0,\ldots,K$
6:         Feed the scaled loss function $f_{t,i}(w) = p_{t,i}\ell(w,y_t)$ to $\mathcal{A}_i$ (Algorithm 2) and obtain $\boldsymbol{q}_{t+1,i}$;

---

It remains to derive the $i$-th external regret algorithm $\mathcal{A}_i$ that minimizes $\mathrm{REG}_i$. Note that $\mathcal{A}_i$ is required to predict a distribution $\boldsymbol{q}_{t,i}$ over $\mathcal{Z}$ and is subsequently fed a scaled loss function $p_{t,i}\ell(.,y_t)$ at each time $t$. We propose to employ the Exponentially Weighted Online Optimization (EWOO) algorithm along with a novel randomized rounding scheme for $\mathcal{A}_i$ (Algorithm 2).

EWOO was studied by Hazan et al. (2007) for minimizing the regret $\sup_{w\in\mathcal{W}}\sum_{t=1}^{T}f_t(w_t) - f_t(w)$, when $\mathcal{W}$ is a convex set, and the loss functions $f_t$'s are exp-concave. Since the log loss is 1-exp-concave in $p$ over $[0,1]$ ((Cesa-Bianchi and Lugosi, 2006, page 46), $\mathrm{EWOO}_i$ (an instance of EWOO for $\mathcal{A}_i$) with functions $\{f_{t,i}\}_{t=1}^{T}$ defined as $f_{t,i}(w) = p_{t,i}\ell(w,y_t)$ for all $w \in \mathcal{W}$, where $\mathcal{W} = [0,1]$ is a natural choice.

Next, we derive a bound on the regret of $\mathrm{EWOO}_i$. Towards this end, we realize that the scaled log loss $f_{t,i}(w) = p_{t,i}\ell(w,y_t)$ is 1-exp-concave since $\exp(-f_{t,i}(w)) = w^{y_t p_{t,i}}(1-w)^{(1-y_t)p_{t,i}}$ is concave when $p_{t,i} \in [0,1]$. Appealing to (Hazan et al., 2007, Theorem 7), we then obtain the following:

**Lemma 4.** *The regret of Algorithm 3 satisfies* $\sup_{w\in\mathcal{W}}\sum_{t=1}^{T}f_{t,i}(w_{t,i}) - f_{t,i}(w) \leq \log(T+1)$.

Note that at each time $t$, $\mathrm{EWOO}_i$ outputs $w_{t,i} \in [0,1]$, however, $\mathcal{A}_i$ is required to predict a distribution $\boldsymbol{q}_{t,i} \in \Delta_{K+1}$ over $\mathcal{Z}$. Thus, we need to perform a rounding operation that projects the output $w_{t,i}$ of $\mathrm{EWOO}_i$ to a distribution over $\mathcal{Z}$. In Remark 1 in Appendix D, we show that the following two known rounding schemes: (a) rounding $w_{t,i}$ to the nearest $z \in \mathcal{Z}$ and setting $\boldsymbol{q}_{t,i}$ as the corresponding one-hot vector; (b) the rounding procedure proposed by Fishelson et al. (2025), cannot be applied to our setting since they incur a $\Omega(1)$ change in the expected loss $\langle\boldsymbol{q}_{t,i},\boldsymbol{\ell}_t\rangle - \ell(w_{t,i},y_t)$, which is not

---

**Algorithm 2** The $i$-th external regret algorithm ($\mathcal{A}_i$)

1: **for** $t = 1, \ldots, T$
2:   Set $w_{t,i} \in [0,1]$ as the output of EWOO$_i$ (Algorithm 3) at time $t$;
3:   Predict $\boldsymbol{q}_{t,i} = \text{RROUND}^{\log}(w_{t,i})$ (Algorithm 4);
4:   Receive the scaled loss function $f_{t,i}(w) = p_{t,i}\ell(w, y_t)$.

---

**Algorithm 3** Exponentially Weighted Online Optimization (EWOO$_i$) with scaled losses

1: **for** $t = 1, \ldots, T$
2:   Set weights $\mu_{t,i}(w) = \exp\left(-\sum_{\tau=1}^{t-1} f_{\tau,i}(w)\right)$ for all $w \in \mathcal{W}$;
3:   Output $w_{t,i} = \frac{\int_{w \in \mathcal{W}} w\mu_{t,i}(w)dw}{\int_{w \in \mathcal{W}} \mu_{t,i}(w)dw}$.

---

sufficient to achieve the desired regret guarantee. To mitigate the shortcomings of these rounding procedures, we propose a different randomized rounding scheme for the log loss (Algorithm 4) that achieves a $\mathcal{O}\left(\frac{1}{K^2}\right)$ change in the expected loss, as per Lemma 5.

**Lemma 5.** *Let $p \in [0,1]$ and $p^-, p^+ \in \mathcal{Z}$ be neighbouring points in $\mathcal{Z}$ such that $p^- \leq p < p^+$. Let $q$ be the random variable that takes value $p^-$ with probability $\propto \frac{p^+ - p}{p^+(1-p^+)}$ and $p^+$ with probability $\propto \frac{p - p^-}{p^-(1-p^-)}$. Then, for all $y \in \{0,1\}$, we have $\mathbb{E}[\ell(q,y)] - \ell(p,y) = \mathcal{O}\left(\frac{1}{K^2}\right).$*

The high-level idea of the proof is as follows: since the log loss is convex in $p$ (for any $y \in \{0,1\}$), we have $\ell(q,y) - \ell(p,y) \leq \ell'(q,y) \cdot (q-p) = \frac{(q-y)(q-p)}{q(1-q)}$, which is $\frac{p}{q} - 1$ if $y = 1$, and $\frac{1-p}{1-q} - 1$ if $y = 0$. By direct computation of $\mathbb{E}\left[\frac{1}{q}\right]$ and $\mathbb{E}\left[\frac{1}{1-q}\right]$, we show that $\mathbb{E}\left[\frac{p}{q}\right] - 1 = \mathbb{E}\left[\frac{1-p}{1-q}\right] - 1 \leq (p^+ - p^-)^2 \cdot \max\left(\frac{1}{p^-(1-p^-)}, \frac{1}{p^+(1-p^+)}\right) = \mathcal{O}\left(\frac{1}{K^2}\right)$, where the last step follows from a technical result due to Lemmas 6 (Appendix C) and 7 (Appendix D).

Combining everything, we derive the regret guarantee REG$_i$ of $\mathcal{A}_i$ (Algorithm 2). It follows from Lemma 5 that at any time $t$, the distribution $\boldsymbol{q}_{t,i}$ obtained by rounding the prediction $w_{t,i}$ of EWOO$_i$ as per Algorithm 4 satisfies $\langle \boldsymbol{q}_{t,i}, \boldsymbol{\ell}_t \rangle = \ell(w_{t,i}, y_t) + \mathcal{O}(\frac{1}{K^2})$. Multiplying with $p_{t,i}$ and summing over all $t$, we obtain

$$\sum_{t=1}^{T} \left\langle \boldsymbol{q}_{t,i} - \boldsymbol{e}_j, \tilde{\boldsymbol{\ell}}_{t,i} \right\rangle = \sum_{t=1}^{T} p_{t,i}\ell(w_{t,i}, y_t) - \sum_{t=1}^{T} p_{t,i}\ell(z_j, y_t) + \mathcal{O}\left(\frac{\sum_{t=1}^{T} p_{t,i}}{K^2}\right),$$

$$\leq \sup_{w \in \mathcal{W}} \sum_{t=1}^{T} f_{t,i}(w_{t,i}) - f_{t,i}(w) + \mathcal{O}\left(\frac{\sum_{t=1}^{T} p_{t,i}}{K^2}\right) = \mathcal{O}\left(\log T + \frac{\sum_{t=1}^{T} p_{t,i}}{K^2}\right),$$

where the last equality follows from Lemma 4. Therefore, the regret REG$_i$ of $\mathcal{A}_i$ satisfies REG$_i$ = $\mathcal{O}\left(\log T + \frac{1}{K^2}\sum_{t=1}^{T} p_{t,i}\right)$. Summing over all $i$, we obtain

$$\sup_{\sigma: \mathcal{Z} \to \mathcal{Z}} \text{PSReg}_\sigma^\ell \leq \sum_{i=0}^{K} \text{REG}_i = \mathcal{O}\left(K \log T + \frac{1}{K^2}\sum_{i=0}^{K}\sum_{t=1}^{T} p_{t,i}\right) = \mathcal{O}\left(K \log T + \frac{T}{K^2}\right).$$

Finally, it follows from (2) that $\text{PSReg}^\ell = \mathcal{O}\left(K \log T + \frac{T}{K^2}\right) = \mathcal{O}\left(T^{\frac{1}{3}}(\log T)^{\frac{2}{3}}\right)$ on choosing $K = (T/\log T)^{\frac{1}{3}}$. Therefore, we have the main result of this section.

**Theorem 2.** *Choosing $K = (T/\log T)^{\frac{1}{3}}$, Algorithm 1 achieves* $\text{PKLCal} = \mathcal{O}\left(T^{\frac{1}{3}}(\log T)^{\frac{2}{3}}\right)$.

Note that Algorithm 1 requires knowledge of the horizon $T$ to choose the discretization parameter $K$. However, since (pseudo) KL-Calibration is equivalent to (pseudo) swap regret of the log loss, we can use the doubling trick to avoid the requirement of knowing the time horizon. The analysis of doubling trick for swap regret is exactly identical to that for external regret and is deferred to

---

**Algorithm 4** Randomized rounding for log loss ($\mathrm{RROUND}^{\log}$)

---

**Input:** $p \in [0,1]$, **Output:** Probability distribution $\boldsymbol{q} \in \Delta_{K+1}$;
**Scheme:** Let $i \in \{0, \ldots, K-1\}$ be such that $p \in [z_i, z_{i+1})$. Output $\boldsymbol{q} \in \Delta_{K+1}$, where

$$q_i = \frac{1}{D} \cdot \frac{z_{i+1} - p}{z_{i+1}(1 - z_{i+1})}, \quad q_{i+1} = \frac{1}{D} \cdot \frac{p - z_i}{z_i(1 - z_i)}, \quad \text{and} \quad q_j = 0, \ \ \forall j \notin \{i, i+1\}$$

with $D = \frac{p - z_i}{z_i(1 - z_i)} + \frac{z_{i+1} - p}{z_{i+1}(1 - z_{i+1})}$ being the normalizing constant.

---

Cesa-Bianchi and Lugosi (2006). Moreover, as we show in Appendix D, the overall computation cost of Algorithm 1 over $T$ rounds is $\tilde{\mathcal{O}}(T^{\frac{5}{3}} + T \cdot \mathrm{ST})$, where ST is the time required to compute the stationary distribution of $Q_t$, which can be obtained efficiently by the method of power iteration; therefore, Algorithm 1 is efficient. In a similar spirit as Corollary 1, we can show Algorithm 1 achieves the following regret bounds simultaneously. The proof is in Appendix D and for most part follows similar to Corollary 1, except that we prove and utilize the bounds (a) $\mathsf{PCal}_1 \leq \sqrt{T \cdot \mathsf{PCal}_2}$; (b) for any $\ell \in \mathcal{L}$, $\mathsf{PSReg}^\ell \leq 4\mathsf{PCal}_1$.

**Corollary 2.** *Algorithm 1 achieves the following bounds simultaneously:*

$$\mathsf{PKLCal} = \mathcal{O}(T^{\frac{1}{3}}(\log T)^{\frac{2}{3}}), \quad \mathsf{PMsr}_{\mathcal{L}_G} = \mathcal{O}(G \cdot T^{\frac{1}{3}}(\log T)^{\frac{2}{3}}),$$

$$\mathsf{PMsr}_{\mathcal{L}_2} = \mathcal{O}(T^{\frac{1}{3}}(\log T)^{\frac{2}{3}}), \quad \mathsf{PMsr}_{\mathcal{L} \setminus \{\mathcal{L}_G \cup \mathcal{L}_2\}} = \mathcal{O}(T^{\frac{2}{3}}(\log T)^{\frac{1}{3}}).$$

We remark that while we do not have a concrete algorithm for KLCal, in Appendix E, we show that if we only consider $\mathcal{L}_G$, then our Algorithm 1 or the algorithm of Fishelson et al. (2025) already achieves a $\mathcal{O}(G \cdot T^{\frac{1}{3}}(\log T)^{-\frac{1}{3}} \log \frac{T}{\delta})$ high probability bound for $\mathsf{Msr}_{\mathcal{L}_G}$.

## 6   Conclusion and Future Directions

In this paper, we introduced a new stronger notion of calibration called (pseudo) KL-Calibration which not only allows us to recover results for classical (pseudo) $\ell_2$-Calibration, but also obtain simultaneous (pseudo) swap regret guarantees for several important subclasses of proper losses. We also derived the first high probability and in-expectation bounds for $\mathsf{Cal}_2$. Several interesting questions remain, including (1) obtaining an explicit high probability swap regret guarantee for the log loss, similar to Section E; (2) improving the $T^{\frac{2}{3}}$ dependence (e.g., to $\sqrt{T}$ as in Hu and Wu (2024)) for a bounded proper loss in Corollaries 1, 2; and (3) studying KL-Calibration in the offline setting.

## Acknowledgement

We thank Fishelson, Kleinberg, Okoroafor, Paes Leme, Schneider, and Teng for sharing a draft of their paper (Fishelson et al., 2025) with us. HL is supported by NSF award IIS-1943607. SS is supported by NSF CAREER Award CCF-2239265. VS is supported by NSF CAREER Award CCF-2239265 and an Amazon Research Award. This work was done in part while VS was visiting the Simons Institute for the Theory of Computing.

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

# Contents

## A   Additional Related Work

**Simultaneous regret minimization**   Kleinberg et al. (2023) proposed U-Calibration, where the goal is to simultaneously minimize the external regret $\text{REG}^\ell$ for all $\ell \in \mathcal{L}$ and provided an algorithm that achieves U-Calibration error $\text{UCal} := \sup_{\ell \in \mathcal{L}} \text{REG}^\ell = \mathcal{O}(\sqrt{T})$. In the multiclass setting with $K$ classes, Luo et al. (2024) proved that the minimax error is $\Theta(\sqrt{KT})$. With an appropriate post-processing of the predictions, the concept of U-Calibration has also been extended to the contextual setting (referred to as online omniprediction (Garg et al., 2024)). Very recently, Okoroafor et al. (2025) have shown that it is possible to achieve $\tilde{\mathcal{O}}(\sqrt{T})$ omniprediction error for a family of bounded variation loss functions against a hypothesis class $\mathcal{F}$ with bounded complexity, thereby surpassing the limitations of swap omniprediction.

**Weaker notions of calibration**   Understanding the limitations of online calibration, i.e., $\mathsf{Cal}_1 = \mathcal{O}(\sqrt{T})$ is impossible, has led to a recent line of work aimed at studying weaker notions of calibration which are still meaningful for downstream loss minimization tasks, e.g., continuous calibration (Foster and Hart, 2021), U-Calibration (Kleinberg et al., 2023), distance to calibration (Qiao and Zheng, 2024; Arunachaleswaran et al., 2025). Particularly, the last two works considered the problem of minimizing the distance to calibration ($\mathsf{CalDist}_1$), defined as the $\ell_1$ distance between the forecaster's vector of predictions and that of the nearest perfectly calibrated predictor, and proposed a non-constructive, constructive proof respectively that there exists an algorithm that achieves $\mathsf{CalDist}_1 = \mathcal{O}(\sqrt{T})$. Since $\mathsf{CalDist}_1 \le \mathsf{Cal}_1 \le \sqrt{T \cdot \mathsf{Cal}_2}$, our Algorithm 1 in fact ensures that $\mathsf{CalDist}_1 = \mathcal{O}(T^{\frac{2}{3}}(\log T)^{-\frac{1}{6}}\sqrt{\log(T/\delta)})$ with probability at least $1 - \delta$, while simultaneously minimizing swap regret for several subclasses of $\mathcal{L}$.

# B   Deferred proofs in Section 3

## B.1   Proof of Proposition 1

*Proof.* For simplicity, we only prove the result for $\mathsf{PSReg}_\sigma^\ell$ since the result for $\mathsf{SReg}_\sigma^\ell$ follows by simply replacing $\mathcal{P}_t(p)$ with $\mathbb{I}[p_t = p]$. We have the following chain of equalities:

$$\mathsf{PSReg}_\sigma^\ell = \sum_{p \in \mathcal{Z}} \sum_{t=1}^T \mathcal{P}_t(p)(\ell(p, y_t) - \ell(\sigma(p), y_t))$$

$$= \sum_{p \in \mathcal{Z}} \sum_{t=1}^T \mathcal{P}_t(p)\left(\ell(p) + \langle \partial\ell(p), y_t - p \rangle - \ell(\sigma(p)) - \langle \partial\ell(\sigma(p)), y_t - \sigma(p) \rangle\right)$$

$$= \sum_{p \in \mathcal{Z}}\left(\sum_{t=1}^T \mathcal{P}_t(p)\right)\left(\ell(p) + \langle \partial\ell(p), \tilde{\rho}_p - p \rangle - \ell(\sigma(p)) - \langle \partial\ell(\sigma(p)), \tilde{\rho}_p - \sigma(p) \rangle\right)$$

$$= \sum_{p \in \mathcal{Z}}\left(\sum_{t=1}^T \mathcal{P}_t(p)\right)\left(\mathsf{BREG}_{-\ell}(\tilde{\rho}_p, p) - \mathsf{BREG}_{-\ell}(\tilde{\rho}_p, \sigma(p))\right),$$

where the second equality follows from Lemma 1, while the final equality follows by adding and subtracting $\ell(\tilde{\rho}_p)$. Taking supremum over $\sigma : [0,1] \to [0,1]$, we obtain

$$\sup_{\sigma:[0,1]\to[0,1]} \mathsf{PSReg}_\sigma^\ell = \sum_{p \in \mathcal{Z}}\left(\sum_{t=1}^T \mathcal{P}_t(p)\right)\left(\mathsf{BREG}_{-\ell}(\tilde{\rho}_p, p) - \inf_{\sigma:[0,1]\to[0,1]} \mathsf{BREG}_{-\ell}(\tilde{\rho}_p, \sigma(p))\right).$$

Next, we realize that $\mathsf{BREG}_\phi(x, y) \ge 0$ since $\phi$ is convex, and the choice of $\sigma(p) = \tilde{\rho}_p$ leads to $\mathsf{BREG}_{-\ell}(\tilde{\rho}_p, \sigma(p)) = 0$. Therefore,

$$\mathsf{PSReg}^\ell = \sum_{p \in \mathcal{Z}}\left(\sum_{t=1}^T \mathcal{P}_t(p)\right)\mathsf{BREG}_{-\ell}(\tilde{\rho}_p, p)$$

which completes the proof. $\qquad\square$

## B.2   Proof of Proposition 2

*Proof.* For simplicity, we only consider the case when $p \le \hat{p}$, since the other case follows exactly similarly. Applying the result of Lemma 2, we obtain

$$\mathsf{BREG}_{-\ell}(\hat{p}, p) = \int_p^{\hat{p}} |\ell''(\mu)|\,(\hat{p} - \mu)d\mu \le c \cdot \int_p^{\hat{p}}\left(\frac{1}{\mu} + \frac{1}{1-\mu}\right) \cdot (\hat{p} - \mu)d\mu,$$

where the inequality follows from Lemma 3. By direct computation, the integral above evaluates to

$$\hat{p} \cdot \int_p^{\hat{p}} \frac{d\mu}{\mu} + (1 - \hat{p}) \cdot \int_p^{\hat{p}} \frac{d\mu}{\mu - 1} = \hat{p} \cdot \log \frac{\hat{p}}{p} + (1 - \hat{p}) \cdot \log \frac{1 - \hat{p}}{1 - p} = \mathsf{KL}(\hat{p}, p).$$

Therefore, we have $\mathsf{BREG}_{-\ell}(\hat{p}, p) \le c \cdot \mathsf{KL}(\hat{p}, p)$, which completes the proof of the first part of the Proposition. The second part follows by combining the result of Proposition 1 with the result obtained above, and taking a supremum over $\ell \in \mathcal{L}_2$. This completes the proof. $\qquad\square$

### B.3 Proof of Proposition 3

*Proof.* For simplicity, we only consider the case when $p \leq \hat{p}$, since the other case follows exactly similarly. Applying the result of Lemma 2, we obtain

$$\mathsf{BREG}_{-\ell}(\hat{p}, p) \leq G \int_p^{\hat{p}} (\hat{p} - \mu) d\mu = G \left( \hat{p}(\hat{p} - p) - \frac{\hat{p}^2 - p^2}{2} \right) = \frac{G}{2} (\hat{p} - p)^2.$$

The case when $\hat{p} \leq p$ follows similarly. Applying the result of Proposition 1, taking a supremum over $\ell \in \mathcal{L}_G$, and bounding $\mathsf{Cal}_2, \mathsf{PCal}_2$ in terms of $\mathsf{KLCal}, \mathsf{PKLCal}$ completes the proof. $\qquad\square$

## C  Deferred proofs in Section 4

### C.1  Proof of Theorem 1

**Theorem 3** (Von-Neumann's Minimax Theorem). *Let $M \in \mathbb{R}^{r \times c}$ for $r, c \in \mathbb{N}$. Then,*

$$\min_{p \in \Delta_r} \max_{q \in \Delta_c} p^\mathsf{T} M q = \max_{q \in \Delta_c} \min_{p \in \Delta_r} p^\mathsf{T} M q.$$

*Proof of Theorem 1.* We prove a stronger statement that the result holds against any adaptive adversary. In the forecasting setup, let $\mathcal{H}_{t-1} = \{p_1, \ldots, p_{t-1}\} \cup \{y_1, \ldots, y_{t-1}\}$ denote the history till time $t$ (exclusive). With complete knowledge about the forecaster's algorithm, an adaptive adversary chooses $y_t$ depending on $\mathcal{H}_{t-1}$. As mentioned in Section 4, we shall consider forecasters that make predictions which belong to the discretization

$$\mathcal{Z} = \{z_1, \ldots, z_{K-1}\}, \text{ where } z_i = \sin^2\left(\frac{\pi i}{2K}\right),$$

and $K \in \mathbb{N}$ is a constant to be specified later. For convinience, we set $z_0 = 0, z_K = 1$, however, $z_0, z_K$ are not included in the discretization. In Lemma 6, we prove some important facts regarding $\mathcal{Z}$ which shall be useful for the subsequent analysis. For a deterministic forecaster, $p_t$ is obtained via a mapping $F_{t-1} : \mathcal{H}_{t-1} \to \mathcal{Z}$. Similarly, for a deterministic adversary, $y_t$ is obtained via a mapping $A_{t-1} : \mathcal{H}_{t-1} \to \{0, 1\}$. Therefore, a deterministic forecaster can be represented by the sequence of mappings $F = (F_1, \ldots, F_T)$, and a deterministic adversary can be represented by the sequence $A = (A_1, \ldots, A_T)$. Given $F, A$, we let $\mathsf{SReg}^\ell(F, A)$ denote the swap regret achieved by executing $F, A$.

Let $\{F\}, \{A\}$ be all possible enumerations of $F, A$ respectively, and $\Delta(\{F\}), \Delta(\{A\})$ denote the set of all distributions over $\{F\}, \{A\}$. Then, $\mathfrak{F} \in \Delta(\{F\}), \mathfrak{A} \in \Delta(\{A\})$ are distributions over $\{F\}, \{A\}$ and represent a randomized forecaster, adversary respectively. Note that $|\{F\}|, |\{A\}| < \infty$, since the domain and range of each map $F_t, A_t$ is finite. Therefore, by Theorem 3, we have

$$\min_{\mathfrak{F} \in \Delta(\{F\})} \max_{\mathfrak{A} \in \Delta(\{A\})} \mathbb{E}_{F \sim \mathfrak{F}, A \sim \mathfrak{A}}[\mathsf{SReg}^\ell(F, A)] = \max_{\mathfrak{A} \in \Delta(\{A\})} \min_{\mathfrak{F} \in \Delta(\{F\})} \mathbb{E}_{F \sim \mathfrak{F}, A \sim \mathfrak{A}}[\mathsf{SReg}^\ell(F, A)].$$
$$(3)$$

For a $v \in \mathbb{R}$, to upper bound the quantity on the right hand side of (3) by $v$, it is sufficient to prove that for any randomized adversary there exists a forecaster $F$ that guarantees that $\mathbb{E}[\mathsf{SReg}^\ell(F, A)] \leq v$. Moreover, swapping the adversary and forecaster allows the forecaster to witness the distribution of $y_t$ before deciding $p_t$. Towards this end, we consider a forecaster $F$ which at time $t$ does the following: (a) it computes $\tilde{p}_t = \mathbb{E}_t[y_t]$; (b) predicts $p_t = \mathrm{argmin}_{z \in \mathcal{Z}} |\tilde{p}_t - z|$.

For each $i \in \{1, \ldots, K-1\}$ and $n \in [T]$, let $n_i(n) := \sum_{t=1}^n \mathbb{I}[p_t = z_i]$. For convinience, we refer to $n_i(T)$ as $n_i$. Fix a $i \in [K-1]$, and define the sequence $X_{1,i}, \ldots, X_{T,i}$ as follows:

$$X_{j,i} := \begin{cases} 0 & \text{if } j > n_i, \\ y_{t_j} - \tilde{p}_{t_j} & \text{if } j \leq n_i. \end{cases}$$

Here $t_j$ denotes the $j$-th time instant when the prediction made is $p_t = z_i$. Observe that the sequence $X_{1,i}, \ldots, X_{T,i}$ is a martingale difference sequence with $|X_{j,i}| \leq 1$ for all $j \in [T]$. In the subsequent steps we obtain a high probability bound on prefix sums of this sequence.

Fix $n \in [T], \mu \in [0, 1], \delta \in [0, 1]$. Applying Lemma 8, we obtain that the following inequality holds with probability at least $1 - \delta$:

$$\left| \sum_{j=1}^{n} X_{j,i} \right| \leq \mu \mathcal{V}_i(n) + \frac{1}{\mu} \log \frac{2}{\delta},$$

where $\mathcal{V}_i(n) = \sum_{j=1}^{\min(n,n_i)} \tilde{p}_{t_j}(1 - \tilde{p}_{t_j})$. To uniformly bound $\mathcal{V}_i(n)$ in terms of $n$, we consider the 2 cases $n \leq n_i$ and $n > n_i$. When $n \leq n_i$, $\mathcal{V}_i(n)$ can be bounded in terms of $z_i$ as follows

$$\mathcal{V}_i(n) = nz_i(1 - z_i) + \sum_{j=1}^{n} \left( \tilde{p}_{t_j}(1 - \tilde{p}_{t_j}) - z_i(1 - z_i) \right)$$

$$= nz_i(1 - z_i) + \sum_{j=1}^{n} (\tilde{p}_{t_j} - z_i) \cdot (1 - \tilde{p}_{t_j} - z_i)$$

$$\leq nz_i(1 - z_i) + \sum_{j=1}^{n} \left| \tilde{p}_{t_j} - z_i \right|$$

$$\leq n \left( z_i(1 - z_i) + \frac{\pi}{2K} \right),$$

where the last inequality follows from Lemma 6. When $n > n_i$, we note that $\mathcal{V}_i(n) = \mathcal{V}_i(n_i) \leq n \left( z_i(1 - z_i) + \frac{\pi}{2K} \right)$, since $n > n_i$. Therefore, with probability at least $1 - \delta$, we have

$$\left| \sum_{j=1}^{n} X_{j,i} \right| \leq \mu n \left( z_i(1 - z_i) + \frac{\pi}{2K} \right) + \frac{1}{\mu} \log \frac{2}{\delta}.$$

Minimizing the bound above with respect to $\mu \in [0, 1]$, we obtain

$$\left| \sum_{j=1}^{n} X_{j,i} \right| \leq \begin{cases} 2\sqrt{n \left( z_i(1 - z_i) + \frac{\pi}{2K} \right) \log \frac{2}{\delta}} & \text{if } n \geq \frac{\log \frac{2}{\delta}}{z_i(1-z_i) + \frac{\pi}{2K}}, \\ n \left( z_i(1 - z_i) + \frac{\pi}{2K} \right) + \log \frac{2}{\delta} & \text{otherwise.} \end{cases}$$

Note that when $n < \frac{\log \frac{2}{\delta}}{z_i(1-z_i) + \frac{\pi}{2K}}$, we can simply bound $n \left( z_i(1 - z_i) + \frac{\pi}{2K} \right) + \log \frac{2}{\delta} < 2 \log \frac{2}{\delta}$. The bounds obtained for both cases can be combined into the following single bound:

$$\left| \sum_{j=1}^{n} X_{j,i} \right| \leq 2\sqrt{\log \frac{2}{\delta}} \cdot \max \left( \sqrt{n \left( z_i(1 - z_i) + \frac{\pi}{2K} \right)}, \sqrt{\log \frac{2}{\delta}} \right),$$

which holds with probability at least $1 - \delta$. Taking a union bound, we obtain that $\left| \sum_{j=1}^{n} X_{j,i} \right| \leq 2\sqrt{\log \frac{2}{\delta}} \cdot \max \left( \sqrt{n \left( z_i(1 - z_i) + \frac{\pi}{2K} \right)}, \sqrt{\log \frac{2}{\delta}} \right)$ holds simultaneously for all $i \in [K - 1], n \in [T]$ with probability at least $1 - (K - 1)T\delta \geq 1 - KT\delta$. In particular, setting $n = n_i$, we obtain that

$$\left| \sum_{j=1}^{n_i} X_{j,i} \right| \leq 2\sqrt{\log \frac{2}{\delta}} \cdot \max \left( \sqrt{n_i \left( z_i(1 - z_i) + \frac{\pi}{2K} \right)}, \sqrt{\log \frac{2}{\delta}} \right) \tag{4}$$

holds for all $i \in [K - 1]$ with probabilty at least $1 - K\delta T$. Equipped with this bound, in the following steps we obtain a high probabilty bound on $\mathsf{SReg}^\ell(F, A)$. This shall be used to bound $\mathbb{E}[\mathsf{SReg}^\ell(F, A)]$ eventually.

We begin by bounding the quantity $|z_i - \rho_i|$, which shall be used to obtain the high probability bound on $\mathsf{SReg}^\ell(F, A)$. We proceed as

$$|z_i - \rho_i| = \frac{1}{n_i} \left| \sum_{t=1}^{T} \mathbb{I}[p_t = z_i](z_i - y_t) \right|$$

$$\leq \frac{1}{n_i} \left( \left| \sum_{t=1}^{T} \mathbb{I}[p_t = z_i](z_i - \tilde{p}_t) \right| + \left| \sum_{t=1}^{T} \mathbb{I}[p_t = z_i](\tilde{p}_t - y_t) \right| \right)$$

$$\leq \max(d_i, d_{i+1}) + \frac{1}{n_i} \left| \sum_{j=1}^{n_i} X_{j,i} \right|,$$

where for each $i \in [K]$, we define $d_i := z_i - z_{i-1}$. The first inequality above follows from the Triangle inequality; the second inequality is because, if $p_t = z_i$, we must have $\tilde{p}_t \in [z_0, \frac{z_1+z_2}{2}]$ if $i = 1$, $\tilde{p}_t \in \left[ \frac{z_{i-1}+z_i}{2}, \frac{z_i+z_{i+1}}{2} \right]$ if $2 \leq i \leq K-2$, and $\tilde{p}_t \in \left[ \frac{z_{K-2}+z_{K-1}}{2}, 1 \right]$ if $i = K-1$, therefore, $|\tilde{p}_t - p_t| \leq \max(d_i, d_{i+1})$. For each $i \in [K-1]$, let $t_i := \frac{\log \frac{2}{\delta}}{z_i(1-z_i)+\frac{\pi}{2K}}$. Next, we write $\mathsf{SReg}^\ell(F, A)$ as

$$\mathsf{SReg}^\ell(F, A) = \underbrace{\sum_{i \in \mathcal{I}} n_i \mathsf{KL}(\rho_i, z_i)}_{\text{Term I}} + \underbrace{\sum_{i \in \bar{\mathcal{I}}} n_i \mathsf{KL}(\rho_i, z_i)}_{\text{Term II}},$$

where $\mathcal{I} := \{ i \in [K-1]; n_i < t_i \}$, and bound Term I, II individually. We begin by bounding Term II in the following manner:

$$\text{Term II} \leq \sum_{i \in \bar{\mathcal{I}}} n_i \chi^2(\rho_i, z_i)$$

$$= \sum_{i \in \bar{\mathcal{I}}} n_i \left( \frac{(\rho_i - z_i)^2}{z_i} + \frac{(\rho_i - z_i)^2}{1 - z_i} \right)$$

$$= \sum_{i \in \bar{\mathcal{I}}} \frac{n_i(\rho_i - z_i)^2}{z_i(1 - z_i)}$$

$$\leq \sum_{i \in \bar{\mathcal{I}}} \frac{2n_i}{z_i(1 - z_i)} \left( (\max(d_i, d_{i+1}))^2 + \left( \frac{1}{n_i} \left| \sum_{j=1}^{n_i} X_{j,i} \right| \right)^2 \right)$$

$$\leq \sum_{i \in \bar{\mathcal{I}}} 2n_i \cdot \frac{(\max(d_i, d_{i+1}))^2}{z_i(1 - z_i)} + 8 \log \frac{2}{\delta} \cdot \left( \sum_{i \in \bar{\mathcal{I}}} \left( \frac{\pi}{2K} \cdot \frac{1}{z_i(1 - z_i)} + 1 \right) \right)$$

$$= \mathcal{O}\left( \frac{T}{K^2} \right) + \mathcal{O}\left( K \log \frac{1}{\delta} \right),$$

where the first inequality follows since $\mathsf{KL}(\rho_i, z_i) \leq \chi^2(\rho_i, z_i)$; the second inequality follows from the bound on $|z_i - \rho_i|$ established above, and since $(a+b)^2 \leq 2a^2 + 2b^2$; the third inequality follows from (4); the final equality follows from Lemma 6, particularly, we use the bounds $\frac{(\max(d_i, d_{i+1}))^2}{z_i(1-z_i)} = \mathcal{O}(\frac{1}{K^2})$ and $\sum_{i=1}^{K-1} \frac{1}{z_i(1-z_i)} = \mathcal{O}(K^2)$. To bound Term I, we first note from the proof of Proposition 2 that

$$n_i \mathsf{KL}(\rho_i, z_i) = \sup_{\sigma:[0,1]\to[0,1]} \sum_{t=1}^{T} \mathbb{I}[p_t = z_i](\ell(p_t, y_t) - \ell(\sigma(p_t), y_t)) \leq n_i \log \frac{1}{\sin^2 \frac{\pi}{2K}},$$

where the last inequality is because for the rounds where $p_t = z_i$, we have

$$\ell(p_t, y_t) \leq \max \left( \log \frac{1}{z_i}, \log \frac{1}{1 - z_i} \right) \leq \max \left( \log \frac{1}{\sin^2 \frac{\pi}{2K}}, \log \frac{1}{1 - \cos^2 \frac{\pi}{2K}} \right) = \mathcal{O}(\log K).$$

(5)

Moreover, repeating the exact same steps done to bound Term II above, we can also bound $n_i \mathsf{KL}(\rho_i, z_i)$ as

$$n_i \mathsf{KL}(\rho_i, z_i) \leq \frac{2n_i}{z_i(1 - z_i)} \left( (\max(d_i, d_{i+1}))^2 + \left( \frac{1}{n_i} \left| \sum_{j=1}^{n_i} X_{j,i} \right| \right)^2 \right)$$

$$= \mathcal{O}\left( \frac{n_i}{K^2} \right) + 8 \left( \log \frac{2}{\delta} \right)^2 \cdot \frac{1}{n_i z_i(1 - z_i)}$$

$$= \mathcal{O}\left( \frac{n_i}{K^2} + \left( \log \frac{1}{\delta} \right)^2 \cdot \frac{1}{n_i z_i(1 - z_i)} \right),$$

where the first equality follows from Lemma 6 and (4). Taking minimum of the two bounds obtained above, we obtain

$$n_i \mathsf{KL}(\rho_i, z_i) = \mathcal{O}\left( \min\left( n_i \log K, \frac{n_i}{K^2} + \left( \log \frac{1}{\delta} \right)^2 \cdot \frac{1}{n_i z_i(1 - z_i)} \right) \right)$$

$$= \mathcal{O}\left( \frac{n_i}{K^2} + \min\left( n_i \log K, \left( \log \frac{1}{\delta} \right)^2 \cdot \frac{1}{n_i z_i(1 - z_i)} \right) \right)$$

$$= \mathcal{O}\left( \frac{n_i}{K^2} + \sqrt{\log K} \log \frac{1}{\delta} \cdot \frac{1}{\sqrt{z_i(1 - z_i)}} \right),$$

where the final inequality follows since for a fixed $a > 0$, $\min(x, \frac{a}{x}) \leq \sqrt{a}$ holds for all $x \in \mathbb{R}$. Summing over $i \in \mathcal{I}$, we obtain the following bound on Term I:

$$\text{Term I} = \mathcal{O}\left( \frac{1}{K^2} \sum_{i \in \mathcal{I}} n_i + \sqrt{\log K} \log \frac{1}{\delta} \cdot \sum_{i \in \mathcal{I}} \frac{1}{\sqrt{z_i(1 - z_i)}} \right) = \mathcal{O}\left( \frac{T}{K^2} + K(\log K)^{\frac{3}{2}} \log \frac{1}{\delta} \right),$$

where the last equality follows from Lemma 6, particularly, $\sum_{i=1}^{K-1} \frac{1}{\sqrt{z_i(1-z_i)}} = \mathcal{O}(K \log K)$. Summarizing, we have shown that

$$\text{Term I} = \mathcal{O}\left( \frac{T}{K^2} + K(\log K)^{\frac{3}{2}} \log \frac{1}{\delta} \right), \quad \text{Term II} = \mathcal{O}\left( \frac{T}{K^2} + K \log \frac{1}{\delta} \right)$$

hold simultaneously with probability at least $1 - KT\delta$. Therefore,

$$\mathsf{SReg}^\ell(F, A) = \mathcal{O}\left( \frac{T}{K^2} + K(\log K)^{\frac{3}{2}} \log \frac{1}{\delta} \right) \tag{6}$$

with probability at least $1 - KT\delta$. To bound $\mathbb{E}[\mathsf{SReg}^\ell(F, A)]$, we let $\mathcal{E}$ be the event in (6). Therefore,

$$\mathbb{E}[\mathsf{SReg}^\ell(F, A)] = \mathbb{E}[\mathsf{SReg}^\ell(F, A)|\mathcal{E}] \cdot \mathbb{P}(\mathcal{E}) + \mathbb{E}[\mathsf{SReg}^\ell(F, A)|\bar{\mathcal{E}}] \cdot \mathbb{P}(\bar{\mathcal{E}})$$

$$= \mathcal{O}\left( \frac{T}{K^2} + K(\log K)^{\frac{3}{2}} \log \frac{1}{\delta} + (K \log K)T^2\delta \right)$$

$$= \mathcal{O}\left( \frac{T}{K^2} + K(\log K)^{\frac{3}{2}} \log T + K \log K \right)$$

$$= \mathcal{O}(T^{\frac{1}{3}}(\log T)^{\frac{5}{3}}),$$

where the second equality follows by using the high probability bound on $\mathsf{SReg}^\ell(F, A)$ obtained in (6), and bounding $\mathbb{E}[\mathsf{SReg}^\ell(F, A)|\bar{\mathcal{E}}] = \mathcal{O}(T \log K)$, which follows from (5); the third equality follows by choosing $\delta = \frac{1}{T^2}$; the final equality follows by choosing $K = \frac{T^{\frac{1}{3}}}{(\log T)^{\frac{5}{6}}}$. This completes the proof. □

**Lemma 6.** *Fix a $k \in \mathbb{N}$. Let $\{z_i\}_{i=0}^K$ be a sequence where $z_0 = 1, z_i = \sin^2\left(\frac{\pi i}{2K}\right)$ for $i = 1, \ldots, K-1$, and $z_K = 1$. For each $i = 1, \ldots, K$, define $d_i := z_i - z_{i-1}$. Then, the following holds: (a) $d_i \leq \frac{\pi}{2K}$ for all $i \in [K]$; (b) $\frac{\max^2(d_i, d_{i+1})}{z_i(1-z_i)} = \mathcal{O}\left(\frac{1}{K^2}\right)$; (c) $\sum_{i=1}^{K-1} \frac{1}{z_i(1-z_i)} = \mathcal{O}(K^2)$; and (d) $\sum_{i=1}^{K-1} \frac{1}{\sqrt{z_i(1-z_i)}} = \mathcal{O}(K \log K)$.*

*Proof.* By direct computation, we have

$$z_i - z_{i-1} = \sin^2 \frac{\pi i}{2K} - \sin^2 \frac{\pi(i-1)}{2K} = \frac{\cos \frac{\pi(i-1)}{K} - \cos \frac{\pi i}{K}}{2} = \sin \frac{\pi}{2K} \sin \left( \frac{\pi}{K} \left( i - \frac{1}{2} \right) \right),$$

(7)

where the second equality follows from the identity $\sin^2 \theta = \frac{1 - \cos 2\theta}{2}$, while the last equality follows from the identity $\cos \alpha - \cos \beta = 2 \sin \frac{\alpha + \beta}{2} \sin \frac{\beta - \alpha}{2}$. Since $\sin \theta \le \theta$ for all $\theta \in \mathbb{R}$, and bounding $\sin \theta \le 1$, we obtain $z_i - z_{i-1} \le \frac{\pi}{2K}$, which completes the proof for the first part of the lemma.

For the second part, we note that

$$\frac{\max^2(d_i, d_{i+1})}{z_i(1 - z_i)} = \frac{\max^2(d_i, d_{i+1})}{\sin^2 \frac{\pi i}{2K} \cos^2 \frac{\pi i}{2K}} = 4 \cdot \frac{\max^2(d_i, d_{i+1})}{\sin^2 \frac{\pi i}{K}},$$

where the second equality follows from the identity $\sin 2\theta = 2 \sin \theta \cos \theta$. It follows from (7) that

$$\max(d_i, d_{i+1}) = \sin \frac{\pi}{2K} \cdot \max \left( \sin \left( \frac{\pi}{K} \left( i - \frac{1}{2} \right) \right), \sin \left( \frac{\pi}{K} \left( i + \frac{1}{2} \right) \right) \right).$$

For simplicity, we assume that $K$ is odd, although a similar treatment can be done for even $K$. Let $1 \le i \le \frac{K-1}{2}$. Then, $\max(d_i, d_{i+1}) = \sin \frac{\pi}{2K} \sin \left( \frac{\pi}{K} \left( i + \frac{1}{2} \right) \right)$. Observe that

$$\frac{\sin \left( \frac{\pi}{K} \left( i + \frac{1}{2} \right) \right)}{\sin \frac{\pi i}{K}} = \frac{\sin \frac{\pi i}{K} \cos \frac{\pi}{2K} + \cos \frac{\pi i}{K} \sin \frac{\pi}{2K}}{\sin \frac{\pi i}{K}} = \cos \frac{\pi}{2K} + \cot \frac{\pi i}{K} \sin \frac{\pi}{2K} \le 1 + \frac{\sin \frac{\pi}{2K}}{\sin \frac{\pi}{K}},$$

where the first equality follows from the identity $\sin(\alpha + \beta) = \sin \alpha \cos \beta + \cos \alpha \sin \beta$, while the inequality follows by noting that $\cot \frac{\pi i}{K} \le \cot \frac{\pi}{K}$ for all $1 \le i \le \frac{K-1}{2}$. Finally, since $\frac{\sin \frac{\pi}{2K}}{\sin \frac{\pi}{K}} = \frac{1}{2 \cos \frac{\pi}{2K}} = \mathcal{O}(1)$, we obtain $\frac{\max^2(d_i, d_{i+1})}{z_i(1-z_i)} = \mathcal{O}(\sin^2 \frac{\pi}{2K}) = \mathcal{O}(\frac{1}{K^2})$. Next, we consider the case when $\frac{K+1}{2} \le i \le K - 1$. Then, $\max(d_i, d_{i+1}) = \sin \frac{\pi}{2K} \sin \left( \frac{\pi}{K} \left( i - \frac{1}{2} \right) \right)$. Repeating a similar analysis as before, we obtain

$$\frac{\sin \left( \frac{\pi}{K} \left( i - \frac{1}{2} \right) \right)}{\sin \frac{\pi i}{K}} = \frac{\sin \frac{\pi i}{K} \cos \frac{\pi}{2K} - \cos \frac{\pi i}{K} \sin \frac{\pi}{2K}}{\sin \frac{\pi i}{K}} = \cos \frac{\pi}{2K} - \cot \frac{\pi i}{K} \sin \frac{\pi}{2K} \le 1 + \frac{\sin \frac{\pi}{2K}}{\sin \frac{\pi}{K}},$$

which is $\mathcal{O}(1)$ as claimed earlier. Therefore, $\frac{\max^2(d_i, d_{i+1})}{z_i(1-z_i)} = \mathcal{O}(\frac{1}{K^2})$. Combining both the cases completes the proof of (b) above.

For (c), similar to (b), we assume for simplicity that $K$ is odd. Then,

$$\sum_{i=1}^{K-1} \frac{1}{z_i(1 - z_i)} = 4 \sum_{i=1}^{K-1} \frac{1}{\sin^2 \frac{\pi i}{K}} = 8 \sum_{i=1}^{\frac{K-1}{2}} \frac{1}{\sin^2 \frac{\pi i}{K}},$$

and the summation $\sum_{i=1}^{\frac{K-1}{2}} \frac{1}{\sin^2 \frac{\pi i}{K}}$ can be bounded in the following manner:

$$\sum_{i=1}^{\frac{K-1}{2}} \frac{1}{\sin^2 \frac{\pi i}{K}} \le \left( \frac{1}{\sin^2 \frac{\pi}{K}} + \int_1^{\frac{K-1}{2}} \frac{1}{\sin^2 \frac{\pi \nu}{K}} d\nu \right)$$

$$\le \left( \frac{1}{\sin^2 \frac{\pi}{K}} + \int_1^{\frac{K}{2}} \frac{1}{\sin^2 \frac{\pi \nu}{K}} d\nu \right)$$

$$= \left( \frac{1}{\sin^2 \frac{\pi}{K}} + \frac{K}{\pi} \int_{\frac{\pi}{K}}^{\frac{\pi}{2}} \frac{1}{\sin^2 \nu} d\nu \right)$$

$$= \left( \frac{1}{\sin^2 \frac{\pi}{K}} + \frac{K}{\pi} \cot \frac{\pi}{K} \right) = \mathcal{O}(K^2).$$

This completes the proof for (c). Repeating the exact same steps as (c) proves (d). We include the full proof for completeness. Observe that

$$\sum_{i=1}^{K-1} \frac{1}{\sqrt{z_i(1-z_i)}} = 2\sum_{i=1}^{K-1} \frac{1}{\sin\frac{\pi i}{K}} = 4\sum_{i=1}^{\frac{K-1}{2}} \frac{1}{\sin\frac{\pi i}{K}},$$

and the summation $\sum_{i=1}^{\frac{K-1}{2}} \frac{1}{\sin\frac{\pi i}{K}}$ can be bounded in the following manner:

$$\sum_{i=1}^{\frac{K-1}{2}} \frac{1}{\sin\frac{\pi i}{K}} \le \frac{1}{\sin\frac{\pi}{K}} + \int_1^{\frac{K-1}{2}} \frac{1}{\sin\frac{\pi\nu}{K}}d\nu \le \frac{1}{\sin\frac{\pi}{K}} + \int_1^{\frac{K}{2}} \frac{1}{\sin\frac{\pi\nu}{K}}d\nu = \frac{1}{\sin\frac{\pi}{K}} + \frac{K}{\pi}\int_{\frac{\pi}{K}}^{\frac{\pi}{2}} \frac{1}{\sin\nu}d\nu.$$

The integral above evaluates to $\log\left(\csc\frac{\pi}{K} + \cot\frac{\pi}{K}\right)$. Therefore, we have that

$$\sum_{i=1}^{K-1} \frac{1}{\sqrt{z_i(1-z_i)}} \le 4\left(\csc\frac{\pi}{K} + \frac{K}{\pi}\log\left(\csc\frac{\pi}{K} + \cot\frac{\pi}{K}\right)\right) = \mathcal{O}(K\log K).$$

This completes the proof. $\qquad\square$

### C.2 Proof of Corollary 1

*Proof.* Let $\mathcal{A}$ be the algorithm guaranteed by Theorem 1. By Pinsker's inequality, we get that $\mathcal{A}$ guarantees $\mathbb{E}[\mathsf{Cal}_2] = \mathcal{O}(T^{\frac{1}{3}}(\log T)^{\frac{5}{3}})$. Moreover, since $\mathsf{Cal}_1 \le \sqrt{T \cdot \mathsf{Cal}_2}$ (Kleinberg et al., 2023, Lemma 13), by Jensen's inequality we have $\mathbb{E}[\mathsf{Cal}_1] \le \sqrt{T \cdot \mathbb{E}[\mathsf{Cal}_2]} = \mathcal{O}(T^{\frac{1}{3}}(\log T)^{\frac{5}{6}})$. Next, (Kleinberg et al., 2023, Theorem 12) states that for any proper loss $\ell$, we have $\mathsf{SReg}^\ell \le 4\mathsf{Cal}_1$. Therefore, $\mathbb{E}[\mathsf{SReg}^\ell] \le 4\mathbb{E}[\mathsf{Cal}_1] = \mathcal{O}(T^{\frac{2}{3}}(\log T)^{\frac{5}{6}})$. Combining this with the result of Proposition 2, 3 completes the proof. $\qquad\square$

## D  Deferred proofs and discussion in Section 5

### D.1 Computational cost of Algorithm 1

The cost of Algorithm 1 at every time step is at most $\mathcal{O}\left(K^2 + \text{INT} + \text{ST}\right)$, where ST is the time required to compute the stationary distribution of $Q_t$ and INT denotes the computation required for evaluating the integral $\frac{\int_0^1 w\mu_{t,i}(w)dw}{\int_0^1 \mu_{t,i}(w)dw}$ in line 3 of Algorithm 3; the $\mathcal{O}(K^2)$ cost is incurred in forming the matrix $Q_t$, and all other operations in Algorithm 1 can be carried out in time that is no worse than $\mathcal{O}(K^2)$. For ST, the stationary distribution of $Q_t$ can be computed by the method of power iteration; notably, each iteration shall incur cost $\mathcal{O}(\text{nnz}(Q_t))$, where $\text{nnz}(Q_t)$ represents the number of non-zero entries in $Q_t$. Since each column of $Q_t$ has at most two non-zero entries (Algorithm 4 randomizes over two adjacent points in the discretization), $\text{nnz}(Q_t) = \Theta(K)$. For INT, the integral is over $[0,1]$ and has a closed-form expression in terms of the gamma function $\Gamma(z) := \int_0^1 \exp(-t)t^{z-1}dt$ as derived below. Recall that

$$f_{\tau,i}(w) = p_{\tau,i}\ell(w,y_\tau) = -p_{\tau,i}\left(y_\tau\log w + (1-y_\tau)\log(1-w)\right)$$
$$= \log\left(w^{-y_\tau p_{\tau,i}}(1-w)^{-p_{\tau,i}(1-y_\tau)}\right).$$

Therefore, $\mu_{t,i}(w) = \exp\left(-\sum_{\tau=1}^{t-1} f_{\tau,i}(w)\right) = w^{\sum_{\tau=1}^{t-1} y_\tau p_{\tau,i}}(1-w)^{\sum_{\tau=1}^{t-1} p_{\tau,i}(1-y_\tau)}$. For convenience, let $\gamma := \sum_{\tau=1}^{t-1} y_\tau p_{\tau,i}, \delta := \sum_{\tau=1}^{t-1} p_{\tau,i}(1-y_\tau)$. Then, $\int_0^1 \mu_{t,i}(w)dw = \int_0^1 w^\gamma(1-w)^\delta dw = \mathrm{B}(\gamma+1,\delta+1)$, where $\mathrm{B}(z_1,z_2)$ denotes the beta function, defined as $\mathrm{B}(z_1,z_2) := \int_0^1 t^{z_1-1}(1-t)^{z_2-1}dt$. Since $\mathrm{B}(z_1,z_2) = \frac{\Gamma(z_1)\Gamma(z_2)}{\Gamma(z_1+z_2)}$ for all $z_1,z_2$ with $z_1,z_2 > 0$, we have

$$\int_0^1 \mu_{t,i}(w)dw = \frac{\Gamma(\gamma+1)\Gamma(\delta+1)}{\Gamma(\gamma+\delta+2)}.$$

Similarly,

$$\int_0^1 w\mu_{t,i}(w)dw = \int_0^1 w^{\gamma+1}(1-w)^\delta dw = \mathrm{B}(\gamma+2,\delta+1) = \frac{\Gamma(\gamma+2)\Gamma(\delta+1)}{\Gamma(\gamma+\delta+3)}.$$

Taking ratio of the two integrals above and using the identity $\Gamma(z+1) = z\Gamma(z)$, which holds for all $z$ with $z > 0$, we obtain

$$\frac{\int_0^1 w\mu_{t,i}(w)dw}{\int_0^1 \mu_{t,i}(w)dw} = \frac{\Gamma(\gamma+2)}{\Gamma(\gamma+1)} \cdot \frac{\Gamma(\gamma+\delta+2)}{\Gamma(\gamma+\delta+3)} = \frac{\gamma+1}{\gamma+\delta+2} = \left(1 + \frac{\delta+1}{\gamma+1}\right)^{-1}.$$

Clearly, at each time $t$, both $\gamma$ and $\delta$ can be computed in $\mathcal{O}(1)$ time using the previously memorized values corresponding to time $t-1$. Therefore, $\mathrm{INT} = \mathcal{O}(1)$. Since $K = \tilde{\Theta}(T^{\frac{1}{3}})$, the overall computation cost over $T$ rounds is $\tilde{\mathcal{O}}(T^{\frac{5}{3}} + T \cdot \mathrm{ST})$.

### D.2 Expected loss of common rounding schemes

We recall the discussion in Section 5: at each time $t$, $\mathrm{EWOO}_i$ outputs $w_{t,i} \in [0,1]$, however, $\mathcal{A}_i$ is required to predict a distribution $\boldsymbol{q}_{t,i} \in \Delta_{K+1}$ over $\mathcal{Z}$. Thus, we need to perform a rounding operation that projects the output $w_{t,i}$ of $\mathrm{EWOO}_i$ to a distribution over $\mathcal{Z}$. In the remark below, we show that the following two known rounding schemes: (a) rounding $w_{t,i}$ to the nearest $z \in \mathcal{Z}$ and setting $\boldsymbol{q}_{t,i}$ as the corresponding one-hot vector; (b) the rounding procedure proposed by Fishelson et al. (2025), cannot be applied to our setting since they incur a $\Omega(1)$ change in the expected loss $\langle \boldsymbol{q}_{t,i}, \boldsymbol{\ell}_t \rangle - \ell(w_{t,i}, y_t)$, which is not sufficient to achieve the desired regret guarantee.

**Remark 1.** *Let $y_t = 1$ and $w_{t,i} = \frac{z_0+z_1}{2}$. The rounding procedure in (a) above ensures that $\boldsymbol{q}_{t,i} = \boldsymbol{e}_0$ with probability one. Therefore, $\langle \boldsymbol{q}_{t,i}, \boldsymbol{\ell}_t \rangle - \ell(w_{t,i}, y_t) = \ell(z_0, 1) - \ell\left(\frac{z_0+z_1}{2}, 1\right) = \log \frac{z_0+z_1}{2z_0}$. Observe that $\frac{z_1}{z_0} = \frac{\sin^2 \frac{\pi}{2K}}{\sin^2 \frac{\pi}{4K}} = 4\cos^2 \frac{\pi}{4K} = 2 + 2\cos \frac{\pi}{2K}$. Therefore, $\langle \boldsymbol{q}_{t,i}, \boldsymbol{\ell}_t \rangle - \ell(w_{t,i}, y_t) = \log\left(\frac{3}{2} + \cos \frac{\pi}{2K}\right) = \Omega(1)$. For the chosen example, the rounding procedure in (b) sets $q_{t,i}(0) = q_{t,i}(1) = \frac{1}{2}$. Thus, $\langle \boldsymbol{q}_{t,i}, \boldsymbol{\ell}_t \rangle - \ell(w_{t,i}, y_t) = \frac{\ell(z_0, 1)+\ell(z_1, 1)}{2} - \ell\left(\frac{z_0+z_1}{2}, 1\right) = \log \frac{z_0+z_1}{2\sqrt{z_0 z_1}} = \log \frac{1+4\cos^2 \frac{\pi}{4K}}{4\cos \frac{\pi}{4K}} = \Omega(1)$.*

### D.3 Proof of Lemma 5

*Proof.* Since the log loss $\ell(p, y)$ is convex in $p$ (for any $y \in \{0, 1\}$), we have

$$\ell(q, y) - \ell(p, y) \le \ell'(q, y) \cdot (q - p) = \frac{(q-y)(q-p)}{q(1-q)} = \begin{cases} \frac{p}{q} - 1 & \text{if } y = 1, \\ \frac{1-p}{1-q} - 1 & \text{if } y = 0. \end{cases} \tag{8}$$

Let $y = 1$. Taking expectation on both sides of (8), we obtain $\mathbb{E}[\ell(q, y)] - \ell(p, y) = \mathbb{E}\left[\frac{p}{q}\right] - 1$. To simplify the expressions involved in the computation of $\mathbb{E}\left[\frac{1}{q}\right]$, we define the normalizing factor $D := \frac{p^+ - p}{p^+(1-p^+)} + \frac{p-p^-}{p^-(1-p^-)}$. By direct computation, we have

$$\mathbb{E}\left[\frac{1}{q}\right] = \frac{1}{D}\left(\frac{p^+ - p}{p^- p^+(1-p^+)} + \frac{p-p^-}{p^- p^+(1-p^-)}\right) = \frac{1}{D} \cdot \frac{(p^+ - p^-)(1-p)}{p^- p^+(1-p^-)(1-p^+)}.$$

Similarly, by direct computation, we obtain

$$D = \frac{p^+ - p}{p^+(1-p^+)} + \frac{p-p^-}{p^-(1-p^-)} = \frac{(p^+ - p^-)(p + p^- p^+ - p(p^- + p^+))}{p^- p^+(1-p^-)(1-p^+)}.$$

Therefore,

$$\mathbb{E}\left[\frac{p}{q}\right] - 1 = \frac{p(1-p)}{p + p^- p^+ - p(p^- + p^+)} - 1 = \frac{(p^+ - p)(p - p^-)}{p + p^- p^+ - p(p^- + p^+)} \le \frac{(p^+ - p^-)^2}{p + p^- p^+ - p(p^- + p^+)}.$$

Next, we let $y = 0$. Taking expectation on both sides of (8), we obtain $\mathbb{E}[\ell(q, y)] - \ell(p, y) = \mathbb{E}\left[\frac{1-p}{1-q}\right] - 1$, thus, we require to bound $\mathbb{E}\left[\frac{1}{1-q}\right]$. Direct computation yields

$$\mathbb{E}\left[\frac{1}{1-q}\right] = \frac{1}{D}\left(\frac{p^+ - p}{p^+(1-p^-)(1-p^+)} + \frac{p - p^-}{p^-(1-p^-)(1-p^+)}\right) = \frac{1}{D} \cdot \frac{p(p^+ - p^-)}{p^- p^+(1-p^-)(1-p^+)}.$$

Substituting the expression for $D$ obtained above, we obtain

$$\mathbb{E}\left[\frac{1-p}{1-q}\right] - 1 = \frac{p(1-p)}{p + p^- p^+ - p(p^- + p^+)} - 1 = \frac{(p^+ - p)(p - p^-)}{p + p^- p^+ - p(p^- + p^+)}$$

$$\leq \frac{(p^+ - p^-)^2}{p + p^- p^+ - p(p^- + p^+)}.$$

Let $f(p) = p + p^- p^+ - p(p^- + p^+)$. Since $f(p)$ is linear in $p$, for any $p \in [p^-, p^+)$, we have $\min(f(p^-), f(p^+)) \leq f(p) \leq \max(f(p^-), f(p^+))$. Since $f(p^-) = p^-(1-p^-)$, $f(p^+) = p^+(1-p^+)$, we obtain

$$\min\left(p^-(1-p^-), p^+(1-p^+)\right) \leq p + p^- p^+ - p(p^- + p^+) \leq \max\left(p^-(1-p^-), p^+(1-p^+)\right)$$

for all $p \in [p^-, p^+)$. Therefore,

$$\mathbb{E}_q[\ell(q, y)] - \ell(p, y) \leq (p^+ - p^-)^2 \cdot \max\left(\frac{1}{p^-(1-p^-)}, \frac{1}{p^+(1-p^+)}\right) = \mathcal{O}\left(\frac{1}{K^2}\right),$$

where the last equality follows from Lemma 7. This completes the proof. $\qquad\square$

**Lemma 7.** *Fix a $k \in \mathbb{N}$. Let $\{z_i\}_{i=0}^K$ be a sequence where $z_0 = \sin^2 \frac{\pi}{4K}, z_i = \sin^2\left(\frac{\pi i}{2K}\right)$ for $i \in [K-1]$, and $z_K = \cos^2 \frac{\pi}{4K}$. For each $i = 1, \dots, K$, define $d_i := z_i - z_{i-1}$. Then, the following holds true for all $i \in [K]$: (a) $\frac{d_i^2}{z_i(1-z_i)} = \mathcal{O}\left(\frac{1}{K^2}\right)$, and (b) $\frac{d_i^2}{z_{i-1}(1-z_{i-1})} = \mathcal{O}\left(\frac{1}{K^2}\right)$.*

*Proof.* It follows from Lemma 6 that (a), (b) hold for all $2 \leq i \leq K-1$. For $i = 1$, since $d_1 \leq z_1 = \sin^2 \frac{\pi}{2K}$, we have

$$\frac{d_1^2}{z_1(1-z_1)} \leq \frac{\sin^4 \frac{\pi}{2K}}{\sin^2 \frac{\pi}{2K} \cos^2 \frac{\pi}{2K}} = \tan^2 \frac{\pi}{2K},$$

which is $\mathcal{O}(\frac{1}{K^2})$ for a large $K$. Similarly, for $i = K$, $d_i = \cos^2 \frac{\pi}{4K} - \cos^2 \frac{\pi}{2K} = \sin^2 \frac{\pi}{2K} - \sin^2 \frac{\pi}{4K} \leq \sin^2 \frac{\pi}{2K}$. Therefore,

$$\frac{d_K^2}{z_K(1-z_K)} \leq \frac{\sin^4 \frac{\pi}{2K}}{\sin^2 \frac{\pi}{4K} \cos^2 \frac{\pi}{4K}} = 4\sin^2 \frac{\pi}{2K} \leq \frac{\pi^2}{K^2},$$

where the equality follows from the identity $\sin 2\theta = 2\sin\theta\cos\theta$. This completes the proof for (a). For (b), when $i = 1$, we have

$$\frac{d_1^2}{z_0(1-z_0)} \leq \frac{\sin^4 \frac{\pi}{2K}}{\sin^2 \frac{\pi}{4K} \cos^2 \frac{\pi}{4K}} = 4\sin^2 \frac{\pi}{2K} \leq \frac{\pi^2}{K^2}.$$

Similarly, when $i = K$, we have

$$\frac{d_K^2}{z_{K-1}(1-z_{K-1})} \leq \frac{\sin^4 \frac{\pi}{2K}}{\sin^2 \frac{\pi}{2K} \cos^2 \frac{\pi}{2K}} = \tan^2 \frac{\pi}{2K},$$

which is $\mathcal{O}(\frac{1}{K^2})$ for a large $K$. This completes the proof. $\qquad\square$

## D.4 Proof of Corollary 2

*Proof.* Since $\mathsf{KLCal} \geq \mathsf{PCal}_2$, Algorithm 1 ensures that $\mathsf{PCal}_2 = \mathcal{O}(T^{\frac{1}{3}}(\log T)^{\frac{2}{3}})$. Next, we show that the $\mathsf{PCal}_1$ satisfies (a) $\mathsf{PCal}_1 \leq \sqrt{T \cdot \mathsf{PCal}_2}$; (b) for any proper loss $\ell$, we have $\mathsf{PSReg}^\ell \leq 4\mathsf{PCal}_1$. The proof is exactly similar to the corresponding variants of (a), (b) above for $\mathsf{Cal}$ as shown by Kleinberg et al. (2023). For (a), applying the Cauchy-Schwartz inequality, we obtain

$$\sum_{p \in \mathcal{Z}} \sum_{t=1}^{T} \mathcal{P}_t(p) |p - \tilde{\rho}_p| \leq \left( \sum_{p \in \mathcal{Z}} \sum_{t=1}^{T} \mathcal{P}_t(p) \right)^{\frac{1}{2}} \left( \sum_{p \in \mathcal{Z}} \sum_{t=1}^{T} \mathcal{P}_t(p)(p - \tilde{\rho}_p)^2 \right)^{\frac{1}{2}} = \sqrt{T \cdot \mathsf{PCal}_2}.$$

Towards showing (b), we first rewrite $\mathsf{PSReg}^\ell = \sum_{p \in \mathcal{Z}} \sum_{t=1}^{T} \mathcal{P}_t(p)\mathsf{BREG}_{-\ell}(\tilde{\rho}_p, p)$, which holds for any proper loss $\ell$ as per Proposition 2. Next, we observe that

$$\mathsf{BREG}_{-\ell}(\tilde{\rho}_p, p) = \ell(p) - \ell(\tilde{\rho}_p) + \partial\ell(p)(\tilde{\rho}_p - p) \leq \partial\ell(\tilde{\rho}_p)(p - \tilde{\rho}_p) + \partial\ell(p)(\tilde{\rho}_p - p)$$
$$\leq 4|p - \tilde{\rho}_p|,$$

where the first inequality follows since $\ell(p)$ is concave; the second inequality follows by noting that $\ell(p, 1) - \ell(p, 0) = \partial\ell(p)$ as per Lemma 1, and since $\ell(p, y) \in [-1, 1]$, we have $|\partial\ell(p)| \leq 2$ for all $p \in [0, 1]$. Substituting the bound on $\mathsf{BREG}_{-\ell}(\tilde{\rho}_p, p)$ obtained above into $\mathsf{PSReg}^\ell$, we obtain $\mathsf{PSReg}^\ell \leq 4\mathsf{PCal}_1$ as desired. Since Algorithm 1 ensures $\mathsf{PCal}_1 = \mathcal{O}(T^{\frac{1}{3}}(\log T)^{\frac{1}{3}})$, we obtain $\mathsf{PSReg}^\ell = \mathcal{O}(T^{\frac{1}{3}}(\log T)^{\frac{1}{3}})$. Combining the above results with Propositions 2, 3 finishes the proof. $\square$

## E High probability bound for maximum swap regret against $\mathcal{L}_G$

While we do not have a concrete algorithm for $\mathsf{KLCal}$, in this section, we show that if we only consider $\mathcal{L}_G$, then our Algorithm 1 or the algorithm of Fishelson et al. (2025) already achieves a $\mathcal{O}(G \cdot T^{\frac{1}{3}}(\log T)^{-\frac{1}{3}} \log \frac{T}{\delta})$ high probability bound for $\mathsf{Msr}_{\mathcal{L}_G}$. To obtain so, we first prove a generic high probability bound that relates $\mathsf{Cal}_2$ with $\mathsf{PCal}_2$. Subsequently, we instantiate our bound with an explicit algorithm for minimizing $\mathsf{PCal}_2$ and use the result of Proposition 3. Our high probability bound in Theorem 4 is independent of the choice of the discretization $\mathcal{Z}$.

**Theorem 4.** *For any algorithm $\mathcal{A}_{\mathsf{Cal}}$, with probability at least $1 - \delta$ over the randomness in $\mathcal{A}_{\mathsf{Cal}}$'s predictions $p_1, \ldots, p_T$, we have $\mathsf{Cal}_2 \leq 6\mathsf{PCal}_2 + 96|\mathcal{Z}| \log \frac{4|\mathcal{Z}|}{\delta}$.*

Our proof of Theorem 4 crucially relies on the following version of Freedman's inequality from Beygelzimer et al. (2011). Refer therein for a proof.

**Lemma 8.** *Let $X_1, \ldots, X_n$ be a martingale difference sequence adapted to the filtration $\mathcal{F}_1 \subseteq \cdots \subseteq \mathcal{F}_n$, where $|X_i| \leq B$ for all $i = 1, \ldots, n$, and $B$ is a fixed constant. Define $\mathcal{V} := \sum_{i=1}^{n} \mathbb{E}[X_i^2 | \mathcal{F}_{i-1}]$. Then, for any fixed $\mu \in \left[0, \frac{1}{B}\right], \delta \in [0, 1]$, with probability at least $1 - \delta$, we have*

$$\left| \sum_{i=1}^{n} X_i \right| \leq \mu\mathcal{V} + \frac{\log \frac{2}{\delta}}{\mu}.$$

*Proof of Theorem 4.* Before discussing the proof, we introduce some notation. Let $\mathcal{Z}$ be enumerated as $\mathcal{Z} = \{z_0, \ldots, z_K\}$, where $K = |\mathcal{Z}| - 1$. Observe that at time $t$, $\mathcal{A}_{\mathsf{Cal}}$ can be equivalently described by the following procedure: (a) it samples $i_t$ from the set $\{0, \ldots, K\}$ with $\mathbb{P}_t(i_t = i) = \mathcal{P}_t(z_i)$, which we write as $\mathcal{P}_{t,i}$ for convenience; (b) forecasts $p_t = z_{i_t}$. Clearly, $\mathbb{I}[p_t = z_i] = \mathbb{I}[i_t = i]$. For simplicity, we denote $\rho_{z_i} = \rho_i$ and $\tilde{\rho}_{z_i} = \tilde{\rho}_i$. Under this notation, $\rho_i, \tilde{\rho}_i$ can be expressed as

$$\rho_i = \frac{\sum_{t=1}^{T} y_t \mathbb{I}[i_t = i]}{\sum_{t=1}^{T} \mathbb{I}[i_t = i]}, \quad \tilde{\rho}_i = \frac{\sum_{t=1}^{T} y_t \mathcal{P}_{t,i}}{\sum_{t=1}^{T} \mathcal{P}_{t,i}}.$$

We begin by bounding $|\rho_i - \tilde{\rho}_i|$ using Lemma 8. Fix a $i \in \{0, \ldots, K\}$ and define the martingale difference sequences $X_t := y_t(\mathcal{P}_{t,i} - \mathbb{I}[i_t = i])$ and $Y_t := \mathcal{P}_{t,i} - \mathbb{I}[i_t = i]$. Observe that $|X_t| \leq$

$1, |Y_t| \leq 1$ for all $t$. Fix a $\mu_i \in [0, 1]$. Applying Lemma 8 to the sequences $X, Y$ and taking a union bound (over $X, Y$), we obtain that with probability at least $1 - \delta$,

$$\left| \sum_{t=1}^{T} y_t (\mathcal{P}_{t,i} - \mathbb{I}[i_t = i]) \right| \leq \mu_i \mathcal{V}_X + \frac{\log \frac{4}{\delta}}{\mu_i}, \quad \left| \sum_{t=1}^{T} \mathcal{P}_{t,i} - \mathbb{I}[i_t = i] \right| \leq \mu_i \mathcal{V}_Y + \frac{\log \frac{4}{\delta}}{\mu_i}, \quad (9)$$

where $\mathcal{V}_X, \mathcal{V}_Y$ are given by

$$\mathcal{V}_X = \sum_{t=1}^{T} \mathbb{E}\left[ X_t^2 | \mathcal{F}_{t-1} \right] = \sum_{t=1}^{T} y_t \cdot \mathcal{P}_{t,i}(1 - \mathcal{P}_{t,i}) \leq \sum_{t=1}^{T} \mathcal{P}_{t,i}, \text{ and}$$

$$\mathcal{V}_Y = \sum_{t=1}^{T} \mathbb{E}\left[ Y_t^2 | \mathcal{F}_{t-1} \right] = \sum_{t=1}^{T} \mathcal{P}_{t,i}(1 - \mathcal{P}_{t,i}) \leq \sum_{t=1}^{T} \mathcal{P}_{t,i}.$$

The upper tail $\rho_i - \tilde{\rho}_i$ can then be bounded in the following manner:

$$\rho_i - \tilde{\rho}_i = \frac{\sum_{t=1}^{T} y_t \mathbb{I}[i_t = i]}{\sum_{t=1}^{T} \mathbb{I}[i_t = i]} - \frac{\sum_{t=1}^{T} y_t \mathcal{P}_{t,i}}{\sum_{t=1}^{T} \mathcal{P}_{t,i}}$$

$$\leq \frac{\sum_{t=1}^{T} y_t \mathbb{I}[i_t = i]}{\sum_{t=1}^{T} \mathbb{I}[i_t = i]} + \frac{\mu_i \sum_{t=1}^{T} \mathcal{P}_{t,i} + \frac{\log \frac{4}{\delta}}{\mu_i} - \sum_{t=1}^{T} y_t \mathbb{I}[i_t = i]}{\sum_{t=1}^{T} \mathcal{P}_{t,i}}$$

$$= \frac{\sum_{t=1}^{T} y_t \mathbb{I}[i_t = i]}{\left( \sum_{t=1}^{T} \mathbb{I}[i_t = i] \right) \left( \sum_{t=1}^{T} \mathcal{P}_{t,i} \right)} \cdot \left( \sum_{t=1}^{T} \mathcal{P}_{t,i} - \mathbb{I}[i_t = i] \right) + \frac{\mu_i \sum_{t=1}^{T} \mathcal{P}_{t,i} + \frac{\log \frac{4}{\delta}}{\mu_i}}{\sum_{t=1}^{T} \mathcal{P}_{t,i}}$$

$$\leq \frac{\sum_{t=1}^{T} y_t \mathbb{I}[i_t = i]}{\left( \sum_{t=1}^{T} \mathbb{I}[i_t = i] \right) \left( \sum_{t=1}^{T} \mathcal{P}_{t,i} \right)} \cdot \left( \mu_i \sum_{t=1}^{T} \mathcal{P}_{t,i} + \frac{\log \frac{4}{\delta}}{\mu_i} \right) + \frac{\mu_i \sum_{t=1}^{T} \mathcal{P}_{t,i} + \frac{\log \frac{4}{\delta}}{\mu_i}}{\sum_{t=1}^{T} \mathcal{P}_{t,i}}$$

$$\leq 2\mu_i + \frac{2 \log \frac{4}{\delta}}{\mu_i \sum_{t=1}^{T} \mathcal{P}_{t,i}},$$

where the first and second inequalities follow from (9), while the last inequality follows by bounding $y_t \mathbb{I}[i_t = i] \leq \mathbb{I}[i_t = i]$. The lower tail can be bounded in an exact same manner as

$$\tilde{\rho}_i - \rho_i = \frac{\sum_{t=1}^{T} y_t \mathcal{P}_{t,i}}{\sum_{t=1}^{T} \mathcal{P}_{t,i}} - \frac{\sum_{t=1}^{T} y_t \mathbb{I}[i_t = i]}{\sum_{t=1}^{T} \mathbb{I}[i_t = i]}$$

$$\leq \frac{\sum_{t=1}^{T} y_t \mathbb{I}[i_t = i] + \mu_i \sum_{t=1}^{T} \mathcal{P}_{t,i} + \frac{\log \frac{4}{\delta}}{\mu_i}}{\sum_{t=1}^{T} \mathcal{P}_{t,i}} - \frac{\sum_{t=1}^{T} y_t \mathbb{I}[i_t = i]}{\sum_{t=1}^{T} \mathbb{I}[i_t = i]}$$

$$= \frac{\sum_{t=1}^{T} y_t \mathbb{I}[i_t = i]}{\left( \sum_{t=1}^{T} \mathcal{P}_{t,i} \right) \left( \sum_{t=1}^{T} \mathbb{I}[i_t = i] \right)} \cdot \left( \sum_{t=1}^{T} \mathbb{I}[i_t = i] - \mathcal{P}_{t,i} \right) + \frac{\mu_i \sum_{t=1}^{T} \mathcal{P}_{t,i} + \frac{\log \frac{4}{\delta}}{\mu_i}}{\sum_{t=1}^{T} \mathcal{P}_{t,i}}$$

$$\leq \frac{\sum_{t=1}^{T} y_t \mathbb{I}[i_t = i]}{\left( \sum_{t=1}^{T} \mathbb{I}[i_t = i] \right) \left( \sum_{t=1}^{T} \mathcal{P}_{t,i} \right)} \cdot \left( \mu_i \sum_{t=1}^{T} \mathcal{P}_{t,i} + \frac{\log \frac{4}{\delta}}{\mu_i} \right) + \frac{\mu_i \sum_{t=1}^{T} \mathcal{P}_{t,i} + \frac{\log \frac{4}{\delta}}{\mu_i}}{\sum_{t=1}^{T} \mathcal{P}_{t,i}}$$

$$\leq 2\mu_i + \frac{2 \log \frac{4}{\delta}}{\mu_i \sum_{t=1}^{T} \mathcal{P}_{t,i}}.$$

Combining both the bounds, we have shown that for a fixed $\mu_i \in [0, 1]$, $|\rho_i - \tilde{\rho}_i| \leq 2\mu_i + \frac{2 \log \frac{4}{\delta}}{\mu_i \sum_{t=1}^{T} \mathcal{P}_{t,i}}$ holds with probability at least $1 - \delta$. Taking a union bound over all $i$, with probability $1 - \delta$, we have

(simultaneously for all $i$)

$$\left| \sum_{t=1}^{T} y_t (\mathcal{P}_{t,i} - \mathbb{I}[i_t = i]) \right| \le \mu_i \sum_{t=1}^{T} \mathcal{P}_{t,i} + \frac{\log \frac{4(K+1)}{\delta}}{\mu_i},$$

$$\left| \sum_{t=1}^{T} \mathcal{P}_{t,i} - \mathbb{I}[i_t = i] \right| \le \mu_i \sum_{t=1}^{T} \mathcal{P}_{t,i} + \frac{\log \frac{4(K+1)}{\delta}}{\mu_i}, \tag{10}$$

$$|\rho_i - \tilde{\rho}_i| \le 2\mu_i + \frac{2 \log \frac{4(K+1)}{\delta}}{\mu_i \sum_{t=1}^{T} \mathcal{P}_{t,i}}. \tag{11}$$

Consider the function $g(\mu) := \mu + \frac{a}{\mu}$, where $a \ge 0$ is a fixed constant. Clearly, $\min_{\mu \in [0,1]} g(\mu) = 2\sqrt{a}$ when $a \le 1$, and $1 + a$ otherwise. Minimizing the bound in (11) with respect to $\mu_i$, we obtain

$$|\rho_i - \tilde{\rho}_i| \le 4 \sqrt{\frac{\log \frac{4(K+1)}{\delta}}{\sum_{t=1}^{T} \mathcal{P}_{t,i}}}, \text{ when } \log \frac{4(K+1)}{\delta} \le \sum_{t=1}^{T} \mathcal{P}_{t,i}.$$

However, when $\log \frac{4(K+1)}{\delta} > \sum_{t=1}^{T} \mathcal{P}_{t,i}$, we obtain that $|\rho_i - \tilde{\rho}_i| \le 2 + \frac{2 \log \frac{4(K+1)}{\delta}}{\sum_{t=1}^{T} \mathcal{P}_{t,i}}$. In particular, when $\sum_{t=1}^{T} \mathcal{P}_{t,i}$ is tiny, which is possible if $\mathcal{A}_{\mathsf{Cal}}$ does not allocate enough probability mass to the index $i$, the bound obtained is large making it much worse than the trivial bound $|\rho_i - \tilde{\rho}_i| \le 1$ which follows since $\rho_i, \tilde{\rho}_i \in [0, 1]$ by definition. Based on this reasoning, we define the set

$$\mathcal{I} := \left\{ i \in \{0, \dots, K\} \text{ s.t. } \log \frac{4(K+1)}{\delta} \le \sum_{t=1}^{T} \mathcal{P}_{t,i} \right\}, \tag{12}$$

and bound $(\rho_i - \tilde{\rho}_i)^2$ as

$$(\rho_i - \tilde{\rho}_i)^2 \le \begin{cases} \frac{16 \log \frac{4(K+1)}{\delta}}{\sum_{t=1}^{T} \mathcal{P}_{t,i}} & \text{if } i \in \mathcal{I}, \\ 1 & \text{otherwise.} \end{cases} \tag{13}$$

Similarly, $\left| \sum_{t=1}^{T} \mathcal{P}_{t,i} - \mathbb{I}[i_t = i] \right|$ can be bounded by substituting the optimal $\mu_i$ obtained above in (10); we obtain

$$\left| \sum_{t=1}^{T} \mathcal{P}_{t,i} - \mathbb{I}[i_t = i] \right| \le \begin{cases} 2\sqrt{\log \frac{4(K+1)}{\delta} \sum_{t=1}^{T} \mathcal{P}_{t,i}} & \text{if } i \in \mathcal{I}, \\ \sum_{t=1}^{T} \mathcal{P}_{t,i} + \log \frac{4(K+1)}{\delta} & \text{otherwise.} \end{cases} \tag{14}$$

Equipped with (13), (14), we proceed to bound $\mathsf{Cal}_2$ in the following manner:

$$\mathsf{Cal}_2 = \sum_{i=0}^{K} \sum_{t=1}^{T} \mathbb{I}[i_t = i] (z_i - \rho_i)^2 \le 2 \sum_{i=0}^{K} \sum_{t=1}^{T} \mathbb{I}[i_t = i] \left( (z_i - \tilde{\rho}_i)^2 + (\rho_i - \tilde{\rho}_i)^2 \right),$$

where the inequality is because $(a + b)^2 \le 2a^2 + 2b^2$ for all $a, b \in \mathbb{R}$. To further bound the term above, we split the summation into two terms $\mathcal{T}_1, \mathcal{T}_2$ defined as

$$\mathcal{T}_1 := \sum_{i \in \mathcal{I}} \sum_{t=1}^{T} \mathbb{I}[i_t = i] \left( (z_i - \tilde{\rho}_i)^2 + (\rho_i - \tilde{\rho}_i)^2 \right),$$

$$\mathcal{T}_2 = \sum_{i \in \bar{\mathcal{I}}} \sum_{t=1}^{T} \mathbb{I}[i_t = i] \left( (z_i - \tilde{\rho}_i)^2 + (\rho_i - \tilde{\rho}_i)^2 \right),$$

and bound $\mathcal{T}_1$ and $\mathcal{T}_2$ individually. We bound $\mathcal{T}_1$ as

$$\mathcal{T}_1 \leq \sum_{i \in \mathcal{I}} \left( \sum_{t=1}^{T} \mathcal{P}_{t,i} + 2\sqrt{\log \frac{4(K+1)}{\delta} \sum_{\tau=1}^{T} \mathcal{P}_{\tau,i}} \right) \left( (z_i - \tilde{\rho}_i)^2 + \frac{16 \log \frac{4(K+1)}{\delta}}{\sum_{\tau=1}^{T} \mathcal{P}_{\tau,i}} \right)$$

$$= \sum_{i \in \mathcal{I}} \sum_{t=1}^{T} \mathcal{P}_{t,i} (z_i - \tilde{\rho}_i)^2 + 16 \log \frac{4(K+1)}{\delta} |\mathcal{I}| +$$

$$2 \sum_{i \in \mathcal{I}} \sqrt{\log \frac{4(K+1)}{\delta} \sum_{\tau=1}^{T} \mathcal{P}_{\tau,i}} \left( (z_i - \tilde{\rho}_i)^2 + \frac{16 \log \frac{4(K+1)}{\delta}}{\sum_{\tau=1}^{T} \mathcal{P}_{\tau,i}} \right)$$

$$\leq \sum_{i \in \mathcal{I}} \sum_{t=1}^{T} \mathcal{P}_{t,i} (z_i - \tilde{\rho}_i)^2 + 16 \log \frac{4(K+1)}{\delta} |\mathcal{I}| + 2 \sum_{i \in \mathcal{I}} \sum_{\tau=1}^{T} \mathcal{P}_{\tau,i} \left( (z_i - \tilde{\rho}_i)^2 + \frac{16 \log \frac{4(K+1)}{\delta}}{\sum_{\tau=1}^{T} \mathcal{P}_{\tau,i}} \right)$$

$$= 3 \sum_{i \in \mathcal{I}} \sum_{t=1}^{T} \mathcal{P}_{t,i} (z_i - \tilde{\rho}_i)^2 + 48 \log \frac{4(K+1)}{\delta} |\mathcal{I}|,$$

where the first inequality follows by substituting the bounds from (13), (14), while the final inequality follows since by the definition of $\mathcal{I}$ in (12), we have $\sqrt{\log \frac{4(K+1)}{\delta} \sum_{\tau=1}^{T} \mathcal{P}_{\tau,i}} \leq \sum_{\tau=1}^{T} \mathcal{P}_{\tau,i}$. Next, we bound $\mathcal{T}_2$ as

$$\mathcal{T}_2 \leq \sum_{i \in \bar{\mathcal{I}}} \left( 2 \sum_{t=1}^{T} \mathcal{P}_{t,i} + \log \frac{4(K+1)}{\delta} \right) \left( (z_i - \tilde{\rho}_i)^2 + 1 \right)$$

$$\leq 2 \sum_{i \in \bar{\mathcal{I}}} \sum_{t=1}^{T} \mathcal{P}_{t,i} (z_i - \tilde{\rho}_i)^2 + 2 \sum_{i \in \bar{\mathcal{I}}} \sum_{t=1}^{T} \mathcal{P}_{t,i} + 2 \log \frac{4(K+1)}{\delta} |\bar{\mathcal{I}}|$$

$$\leq 2 \sum_{i \in \bar{\mathcal{I}}} \sum_{t=1}^{T} \mathcal{P}_{t,i} (z_i - \tilde{\rho}_i)^2 + 4 \log \frac{4(K+1)}{\delta} |\bar{\mathcal{I}}|,$$

where the first inequality follows by substituting the bounds from (13), (14); the second inequality follows by bounding $(z_i - \tilde{\rho}_i)^2 \leq 1$; the final inequality follows from the definition of $\mathcal{I}$ (12). Collecting the bounds on $\mathcal{T}_1$ and $\mathcal{T}_2$, we obtain

$$\mathcal{T}_1 + \mathcal{T}_2 \leq 3 \sum_{i=0}^{K} \sum_{t=1}^{T} \mathcal{P}_{t,i} (z_i - \tilde{\rho}_i)^2 + 48 \log \frac{4(K+1)}{\delta} |\mathcal{I}| + 4 \log \frac{4(K+1)}{\delta} |\bar{\mathcal{I}}|$$

$$\leq 3\mathsf{PCal}_2 + 48(K+1) \log \frac{4(K+1)}{\delta},$$

where the last inequality follows from the definition of $\mathsf{PCal}_2$ and since $|\mathcal{I}| + |\bar{\mathcal{I}}| = K+1$. Since $\mathsf{Cal}_2 \leq 2(\mathcal{T}_1 + \mathcal{T}_2)$, we have shown that

$$\mathsf{Cal}_2 \leq 6\mathsf{PCal}_2 + 96(K+1) \log \frac{4(K+1)}{\delta} \tag{15}$$

with probability at least $1 - \delta$. This completes the proof. $\qquad \square$

Instantiating $\mathcal{A}_{\mathsf{Cal}}$ in Theorem 4, we obtain the following corollary.

**Corollary 3.** *On choosing $K = (T/\log T)^{\frac{1}{3}}$, Algorithm 1 ensures that with probability at least $1 - \delta$ over its internal randomness*

$$\mathsf{Cal}_2 = \mathcal{O}\left( T^{\frac{1}{3}} (\log T)^{-\frac{1}{3}} \log \frac{T}{\delta} \right), \quad \mathsf{Msr}_{\mathcal{L}_G} = \mathcal{O}\left( G \cdot T^{\frac{1}{3}} (\log T)^{-\frac{1}{3}} \log \frac{T}{\delta} \right).$$

*Furthermore, $\mathbb{E}[\mathsf{Cal}_2] = \mathcal{O}(T^{\frac{1}{3}} (\log T)^{\frac{2}{3}}), \mathbb{E}[\mathsf{Msr}_{\mathcal{L}_G}] = \mathcal{O}(G \cdot T^{\frac{1}{3}} (\log T)^{\frac{2}{3}}).$*

*Proof.* Since Algorithm 1 ensures that $\mathsf{PCal}_2 = \mathcal{O}\left(\frac{T}{K^2} + K \log T\right)$ (refer Section 5), we obtain

$$\mathsf{Cal}_2 = \mathcal{O}\left(\frac{T}{K^2} + K \log T + K \log \frac{K}{\delta}\right)$$

with probability at least $1 - \delta$, which is $\mathcal{O}\left(\frac{T^{\frac{1}{3}}}{(\log T)^{\frac{1}{3}}} \log \frac{T}{\delta}\right)$ on substituting $K$. The high probability bound on $\mathsf{Msr}_{\mathcal{L}_G}$ follows since $\mathsf{Msr}_{\mathcal{L}_G} \leq G \cdot \mathsf{Cal}_2$. To bound $\mathbb{E}[\mathsf{Cal}_2]$, we let $\mathcal{E}$ denote the event that $\mathsf{Cal}_2 \leq \Delta$, where $\Delta := 6\mathsf{PCal}_2 + 96(K+1) \log \frac{4(K+1)}{\delta}$. We then have,

$$\mathbb{E}[\mathsf{Cal}_2] = \mathbb{E}[\mathsf{Cal}_2|\mathcal{E}] \cdot \mathbb{P}(\mathcal{E}) + \mathbb{E}[\mathsf{Cal}_2|\bar{\mathcal{E}}] \cdot \mathbb{P}(\bar{\mathcal{E}}) = \mathcal{O}\left(\frac{T}{K^2} + K \log T + K \log \frac{K}{\delta} + \delta \cdot T\right)$$

which is $\mathcal{O}(T^{\frac{1}{3}} (\log T)^{\frac{2}{3}})$ on substituting $\delta = \frac{1}{T}$ and $K$. Note that the second equality above follows since $\mathbb{E}[\mathsf{Cal}_2|\mathcal{E}] \leq \Delta$ and $\mathbb{P}(\mathcal{E}) \leq 1$, $\mathsf{Cal}_2 \leq T$ and $\mathbb{P}(\bar{\mathcal{E}}) < \delta$. Finally, bounding $\mathsf{Msr}_{\mathcal{L}_G} \leq G \cdot \mathsf{Cal}_2$ finishes the proof. $\square$

Instantiating $\mathcal{A}_{\mathsf{Cal}}$ with the algorithm of Fishelson et al. (2025), we also obtain the exact same guarantee as Corollary 3. Compared to Algorithm 1, the algorithm of Fishelson et al. (2025) is more efficient since it uses scaled online gradient descent for the $i$-th external regret algorithm, which is more efficient than $\mathsf{EWOO}_i$. On the contrary, it does not posses the generality of Algorithm 1 towards minimizing $\mathsf{SReg}^\ell$ for all $\ell \in \mathcal{L}_2$ simultaneously.

