# OpenReview forum: "Simultaneous Swap Regret Minimization via KL-Calibration"
_NeurIPS.cc/2025/Conference — NeurIPS 2025 spotlight_

### Official Review · Reviewer_y6ag · 2025-06-25

**Clarity:** 3
**Significance:** 2
**Originality:** 2
**Rating:** 5
**Confidence:** 4

**Summary:**

This work studies online calibration---the problem of making probabilistic predictions on a sequence of $T$ (arbitrary) binary outcomes---with respect to several different calibration measures.

The main contribution is the introduction of the KL-calibration error (and a "pseudo" version thereof). The authors proved that:
- An $\widetilde O(T^{1/3})$ KL-calibration error can be guaranteed.
- An $\widetilde O(T^{1/3})$ pseudo KL-calibration error can be achieved by an explicit algorithm.

Furthermore, the KL-calibration error (resp., the pseudo version) is shown to upper bound the swap regret (resp., the pseudo swap regret) with respect to every proper scoring rule with a univariate form that is either smooth or twice continuously differentiable. As an application, the results above imply an $\widetilde O(T^{1/3})$ bound on the $\ell_2$ calibration error, recovering a recent result of Fishelson, Kleinberg, Okoroafor, Paes Leme, Schneider, and Teng (ALT 2025).

**Questions:**

I don't have any specific questions.

**Ethical Concerns:**

["NO or VERY MINOR ethics concerns only"]

**Final Justification:**

I did not mention any specific questions in my review, so no response/further discussion was necessary. My evaluation of the paper remains positive after reading the other reviews and the authors' response to them.

**Limitations:**

Yes

**Quality:**

3

**Strengths And Weaknesses:**

Strengths:
- Online calibraion is a well-studied problem in learning theory. The newly-introduced KL-calibration is a natural calibration measure that has a nice decision-theoretic interpretation in terms of the log loss.
- More interestingly, KL-calibration simultaneously upper bounds the swap regret with respect to many losses, and enjoys a much faster rate of $T^{1/3}$ compared to the $\sqrt{T}$ rate shown by [Hu and Wu, FOCS'24] (for all bounded proper losses).
- In addition, while the proofs use several prior ideas (such as the minimax argument and the Blum-Mansour reduction), they still involve new ideas such as the non-uniform discretization, which might be of independent interest.
- The writing of the paper is very clear. The authors did a great job in comparing their results to previous ones, as well as adequately crediting the ideas to prior work.

Weaknesses:
- The results for $\ell_2$ calibration---arguably the most well-studied measure to which the results in current paper applie---are a bit incremental: Foster and Vohra (1998), Hart (2022), and Fishelson et al. (2025) have already proved that: (1) An $\widetilde O(T^{1/3})$ $\ell_2$ calibration error can be achieved (inefficiently); (2) An $\widetilde O(T^{1/3})$ pseudo $\ell_2$ calibration error can be achieved (efficiently). So, as far as $\ell_2$ calibration is concerned, it appears that the only new result is the high-probability bound (Appendix E).

Despite the weaknesses outlined above, this paper presents fairly solid results and is a very nice addition to the recent line of work on online calibration---a problem that is of clear interest to the NeurIPS community. I vote for acceptance.

Other comments / writing suggestions:
- Line 74: It might be more clear if it is emphasized that the big-O notations hide factors that depend on the loss $\ell$ (but not $T$), which aligns better with Lines 76-77 for $\mathcal{L}_G$.
- Reference [FKOPST25]: The last name of the fourth author would be "Paes Leme" instead of "Leme". (See the BibTeX file at https://proceedings.mlr.press/v272/fishelson25a.html.)

---

> ### Author Rebuttal · Authors · 2025-07-29
>
> We sincerely thank the reviewer for their positive evaluation of our work. We shall carefully incorporate your writing suggestions in the subsequent revisions.

---

### Official Review · Reviewer_vKYY · 2025-07-03

**Clarity:** 2
**Significance:** 3
**Originality:** 3
**Rating:** 5
**Confidence:** 4

**Summary:**

This paper introduces a new notion of calibration based on KL divergence and shows that it is equivalent to swap regret for the log loss. The KL calibration error recovers recent results for the $\ell_2$ calibration error as a special cases, and extends to a broader class of losses such as the log loss and Tsallis entropy.

**Questions:**

- The paper focuses on binary outcomes; is there any hope to extend the concept of KL Calibration to multiclass outcomes?
- An oblivious adversary is assumed but it is claimed that the results generalize to an adaptive adversary. What needs to change in the analysis? Is it significantly more complicated, despite being technically possible?

**Ethical Concerns:**

["NO or VERY MINOR ethics concerns only"]

**Final Justification:**

It is an obviously very strong paper with strong results. The writing is a bit sloppy but I trust that it can be cleaned up sufficiently before the camera-ready.

**Limitations:**

yes

**Quality:**

3

**Strengths And Weaknesses:**

# Strengths

The paper does a lot of things in what feels like the "right" way; $\ell_p$ calibration has always seemed weird to me because you're measuring distances between probabilities using an $\ell_p$ norm. The KL Calibration instead measures using a KL divergence, which seems much more natural of a way to measure deviation from a comparator distribution.

Similarly, most calibration papers seem to discretize $[0,1]$ using a uniform strategy, whereas this work uses (essentially) the nodes of the Chebyshev polynomials, which get more dense near the boundaries, which again just seems like the "right" way to discretize $[0,1]$ in this setting. I found these two insights very satisfying. Both of these results involve non-trivial extensions to the existing approaches.

# Weaknesses

The algorithm achieving KL Calibration is non-constructive, so only the existence is shown but no concrete algorithm is given (though the authors are upfront about this).

My biggest concern is that the writing is not easy to follow in general and feels quite "drafty". Many of the explanations in the main text are essentially proof sketches but with too many details to be helpful as a high-level overview---It sort of feels like it's simultaneously trying to fill space (too many low-level details in the proof sketches) while also trying to save space (cramming many bulky expressions inline).

In section 4, for example, in Step II it doesn't seem straight-forward how to bound $|z_i-\rho|$ using the displayed math; is the reader supposed to be able to see the connection here? Or are we supposed to take it at face value for now and defer to the appendix for the specifics? The ambiguity about which details the reader is supposed to understand now vs in the appendix makes these sketches frustrating to read.

Similarly, in Step III I recognize all of the terms and can follow what's being said, but at the same time I couldn't summarize most of what it said right after reading it, so it's not really serving its intended purpose as a sketch of ideas or giving useful insights about the techniques.

The full explanations in the appendix, however, generally seemed quite clear to me, so I would be willing to trust that the authors could improve the writing quality between now and the camera-ready.

**Typos etc**
- 269: "The bound above dictates separate consideration of $i\in I$ and $i\in I$"

---

> ### Author Rebuttal · Authors · 2025-07-29
>
> We sincerely thank the reviewer for their positive comments and time in evaluating our paper. We shall carefully incorporate your writing suggestions in the subsequent revisions. Below, we provide responses to all your questions.
>
> > The paper focuses on binary outcomes; is there any hope to extend the concept of KL Calibration to multiclass outcomes?
>
> It is not immediately trivial to extend the concept of KL-Calibration to multiclass outcomes, as we are not familiar with a generalization of the non-uniform discretization to the multiclass setting that still ensures desirable properties. Even more, we are not familiar with an extension of the randomizing rounding procedure for the log loss (utilized to obtain an explicit algorithm for minimizing pseudo KL-Calibration) to the multiclass setting. Moreover, a reason we did not seriously consider the possibility of such an extension to the multiclass setting is that several implications of KL-Calibration towards minimizing swap regret do not necessarily hold. In particular, in the binary setting, the implication $\text{SReg}^\ell = \mathcal{O}(\text{KCal})$ for each $\ell \in \mathcal{L}_2 \cup \mathcal{L}_G$ is derived via Lemma 2, which states that for a proper loss $\ell$, we have  $\text{BREG}(\hat{p}, p) =\int_p^\hat{p} |\ell''(\mu)| \cdot (\hat{p} - \mu) d\mu$;  the proof of the above result is based on a basis characterization of proper losses over binary outcomes in terms of V-shaped losses $\ell_v(p, y) = (v - y) \cdot \text{sign}(p - v)$ --- for each proper loss $\ell$, there exists a non-negative function (specific to $\ell$) $\mu: [0, 1] \to [0, 1]$ satisfying $\int_0 ^ 1 \mu(v) dv \le 2$ and $\ell (p, y) = \int_0^1 \mu(v) \ell_v (p, y) dv$ (ref. Theorem 8 in Kleinberg et al.). However, a similar characterization of proper losses is not known in the multiclass setting, and in fact cannot exist via the impossibility result in Theorem 20 in Kleinberg et al.
>
> Kleinberg, B., Leme, R. P., Schneider, J., & Teng, Y. (2023, July). U-calibration: Forecasting for an unknown agent. In The Thirty Sixth Annual Conference on Learning Theory (pp. 5143-5145). PMLR.
>
> > An oblivious adversary is assumed, but it is claimed that the results generalize to an adaptive adversary. What needs to change in the analysis? Is it significantly more complicated, despite being technically possible?
>
> The only result for which the analysis becomes simpler with an oblivious adversary is the high probability bound $\text{Cal}_2 \le 6 \text{PCal}_2 + 96 |\mathcal{Z}| \log \frac{4 |\mathcal{Z}|}{\delta}$ (Theorem 4 in Appendix E); we use Theorem 4 to derive a high probability bound on the classical $\ell_2$-Calibration ($\text{Cal}_2$). Extending the above result for an adaptive adversary is simple, but requires a more involved application of Freedman's inequality, which is outlined below. All other results in our paper apply as is for an adaptive adversary.
>
> In more detail, Freedman's inequality states that for a martingale difference sequence $X_1, \dots, X_T$, where $|X_t| \le B$ for all $t \in [T]$, we have $|\sum_{t = 1} ^ {T} X_t| \le \mu V_X + \frac{1}{\mu} \log \frac{2}{\delta}$ with probability at least $1 - \delta$, where $V_X = \sum_{t = 1} ^ {T} \mathbb{E}_t[X_t ^ {2}]$, $\mu \in [0, \frac{1}{B}]$ is a fixed constant, and $\delta \in (0, 1)$. When $V_X$ is a deterministic quantity, the upper bound in the display above can be minimized with respect to $\mu \in [0, \frac{1}{B}]$; the optimal $\mu^\star = \min\left(\frac{1}{B}, \sqrt{\frac{\log \frac{2}{\delta}}{V_X}}\right)$.  For the considered martingale difference sequences in Theorem 4 (e.g., one such sequence is $X_t \coloneqq y_t(\mathcal{P}_t(i) - \mathbb{I}[i_t = i]$), the corresponding $V_X$ is deterministic since the adversary is oblivious (note also that our algorithm outputs $\mathcal{P}_t$ in a deterministic manner). However, when $V_X$ is random (which is indeed the case in Lemma 9), we cannot merely substitute $\mu^\star$ in the bound above since $\mu^\star$ is random. To circumvent this difficulty, we can partition the feasible interval for $\mu$ geometrically (e.g., when $B = 2$, we partition to intervals of the form $[\frac{1}{2^{k + 1}}, \frac{1}{2^{k}}]$), apply Freedman's inequality within each interval, and subsequently take a union bound over all intervals. Notably, each interval corresponds to a condition on $V_X$ for which the optimal $\mu$ falls within that interval, and for that interval substituting $\mu = \frac{1}{2^{k}}$ or $\mu = \frac{1}{2^{k + 1}}$ leads to only a lower-order blow-up in the quantity $\mu V_X + \frac{1}{\mu} \log \frac{2}{\delta}$ compared to substituting $\mu^\star$.
>
> We hope that our responses answer your questions. If you have any further questions, please don't hesitate to reach out.

---

> > ### Comment · Reviewer_vKYY · 2025-08-01
> >
> > Great, thank you for the thoughtful responses!

---

### Official Review · Reviewer_24mT · 2025-07-07

**Clarity:** 3
**Significance:** 2
**Originality:** 2
**Rating:** 5
**Confidence:** 3

**Summary:**

This paper studies the problem of online calibrated prediction of Boolean outcomes. They consider (pseudo) KL-Calibration, which they show is equivalent to (pseudo) swap regret for log loss. They further show (pseudo) KL calibration implies (pseudo) L2 calibration and bounded (pseudo) swap regret for any bounded proper loss with a twice differentiable univariate form, as well as bounded swap regret for any proper loss with a G-smooth univariate form. Using their log loss swap regret characterization of KL-calibration, they give a non-constructive proof of the existence of an algorithm with expected KL Calibration error that is $O(T^{1/3}(\log T)^{5/3})$, improving over recent pseudo L2 calibration error bounds by generalizing the class of losses to which the regret bound applies, as well as giving bounds (via non-constructive argument) for expected swap regret for the log loss (rather than pseudo swap regret).

**Questions:**

Please see comments above.

**Ethical Concerns:**

["NO or VERY MINOR ethics concerns only"]

**Final Justification:**

My appraisal of the paper is positive and I am keeping my score for the reasons given in the original review.

**Limitations:**

Yes

**Paper Formatting Concerns:**

No concerns.

**Quality:**

3

**Strengths And Weaknesses:**

The paper studies interesting and important questions about the notions of calibration for which efficient online prediction is feasible, and explores the relationship between several useful notions of (pseudo) swap regret and calibration. The paper makes good progress on these questions by, among other contributions, generalizing recent results to a much broader class of losses.

The paper is well-written and the new techniques used to obtain their results seem like a nice contribution as well.

The authors give some examples of losses to which their results apply, but as a reader, I think having some additional discussion of the sorts of problems that give rise to losses covered by their new results would great improve my ability to contextualize the contribution. Log loss and squared error are familiar, but spherical error not as much, and understanding what we can get from generalizing prior results beyond psuedo l2 calibration error to these other losses would be helpful.
I’d also be interested in seeing some discussion of lower bounds for pseudo swap regret (if there’s anything of note in the literature to discuss).

---

> ### Author Rebuttal · Authors · 2025-07-29
>
> We sincerely thank the reviewer for their positive evaluation of the paper. We shall incorporate your suggestions in the subsequent revisions of our paper.
>
> > I think having some additional discussion of the sorts of problems that give rise to losses covered by their new results would great improve my ability to contextualize the contribution. Log loss and squared error are familiar, but spherical error not as much, and understanding what we can get from generalizing prior results beyond psuedo $\ell_{2}$-calibration error to these other losses would be helpful.
>
> We agree that the spherical loss $\ell(p, y) = -\frac{y p + (1 - y) (1 - p)}{\sqrt{p ^ {2} + (1 - p) ^ {2}}}$ is less familiar than the log loss and the squared loss. However, it has been previously studied in the literature to evaluate the quality of predictions made by a forecaster (ref. Gneiting and Raftery). Furthermore, the spherical loss has a nice geometric interpretation in terms of a cosine similarity (specifically, $\ell(p, y) = \cos(\pi - \theta)$, where $\theta$ is the angle between the vectors $[p, 1 - p]$ and $[y, 1 - y]$), therefore may be useful in problems where angular separation is meaningful. Similarly, other losses mentioned in our paper, e.g., losses induced by the Tsallis entropy have been previously studied in online learning and been used as regularizers in the classical Follow-the-Regularized Leader (FTRL) algorithm (ref. Abernethy et al.).
>
> Gneiting, T. and Raftery, A. E. (2007). Strictly proper scoring rules, prediction, and estimation. Journal of the American statistical Association, 102(477):359–378.
>
> Abernethy, J. D., Lee, C., \& Tewari, A. (2015). Fighting bandits with a new kind of smoothness. Advances in Neural Information Processing Systems, 28.
>
> > I’d also be interested in seeing some discussion of lower bounds for pseudo swap regret (if there’s anything of note in the literature to discuss).
>
> Regarding lower bounds, we would like to mention that the section on related work (line 109 onwards) does mention some lower bounds for $\text{Cal}_1, \text{Cal}_2$ (and thus swap regret of the squared loss). In summary, proving lower bounds for online calibration, swap regret is a significantly challenging problem and obtaining the minimax bound is still open, with recent breakthroughs establishing that $\tilde{\Omega}(T^{0.54389}) \le \mathbb{E}[\text{Cal}_1] \le \tilde{\mathcal{O}}(T^{\frac{2}{3} - \varepsilon})$ (for a sufficiently small $\varepsilon > 0$) and $\tilde{\Omega}(T^{0.08778}) \le \mathbb{E}[\text{Cal}_2] \le \tilde{\mathcal{O}}(T^{\frac{1}{3}})$ (Dagan et al., Fishelson et al.).
>
> We hope that our responses answer your questions. If you have any further questions, please don't hesitate to reach out.

---

> ### Comment · Reviewer_24mT · 2025-08-06
>
> Thank you to the authors for their helpful rebuttal, I will maintain my score.

---

### Official Review · Reviewer_pP17 · 2025-07-08

**Clarity:** 2
**Significance:** 3
**Originality:** 3
**Rating:** 5
**Confidence:** 3

**Summary:**

Calibration measures provide means of quantifying the "goodness" of predictions. Typically, calibration is measured as some combination of squared/absolute bias across different level sets of one's predictions. This paper instead proposes to measure bias at each level set of a predictor using KL divergence, which they refer to as KL-calibration (and a variant that, in a sense, averages over the randomness from the sampling of the forecaster's predictions, which they refer to as pseudo KL-calibration).

Because this calibration measure is stronger than (upper bounds) l2 calibration, the measure is strong enough to provide swap regret guarantees for a broader class of proper losses including log loss; KL calibration ends up being exactly the worst-case swap-regret of log loss. In particular, because KL upper bounds the Bregman divergence of any twice differentiable proper loss, KL calibration also implies bounds on swap regret for every twice-differentiable proper loss.

The authors first follow the usual Hart's min-max theorem approach to showing the existence of a calibrated forecaster to prove a non-constructive rate for actual KL calibration. The authors claim the technical difficulty of the result is deciding how to discretize the action space of the forecaster (which we need for min-max); rather than using the usual uniform bins, the authors implement some sinusoidal grid. The resulting rate is of the order \tilde O(T^{1/3}).

For an explicit forecasting algorithm for pseudo KL calibration, the paper follows the Blum-Mansour reduction using exponential weights and by using a randomized rounding scheme more suited for the non-smooth curvature of log losses. The resulting rate is of the order O(T^{1/3}).

**Questions:**

My impression of the paper is positive so I don't have any questions in particular for the authors, other than just generally the (mild) weaknesses.

**Ethical Concerns:**

["NO or VERY MINOR ethics concerns only"]

**Final Justification:**

I enjoyed the paper and maintain my positive sentiment.

**Limitations:**

Yes

**Quality:**

4

**Strengths And Weaknesses:**

Strengths:

I did not check the correctness of the proofs in the Appendix beyond reviewing the sketches in the main body. Nonetheless, the main technical insights seem to be a combination of the following facts 1) KL gives a natural worst-case bound on log loss and twice differentiable losses, 2) standard approaches to showing calibration rate existence and building swap regret algorithms can be adapted even to non-Lipschitz losses with more careful discretization/rounding. Both facts seem reasonable and 1 in particular is a nice observation that the authors give a nice treatment of in this paper.

Weaknesses:

I don't have many specific concerns/weaknesses with the paper, which I generally find to be a nice set of results. If I were to nitpick, I'd note that the paper is written for a very specific and therefore narrow audience of people who work on calibration. The paper—even just the first 68 lines—are not written to be very friendly to a more general audience (though I'm sure this is just an artifact of squeezing a COLT-size paper into the Neurips format).

I think another thing is that for readers that haven't read the Fichelson et al 2025 paper to define pseudo-calibration more formally (specifically calP) and give a more gentle explanation for what intuition one should walk in with about why relaxing to pseudo-notions of calibration allows for the sqrt T - T^{ 1/3} gap (does the noise in realizing the forecasting's predictions in non-pseudo calibration cause one to naively pay some variance term of order sqrt T? why does such a variance term exist, especially if the forecaster only needs to randomized over two very close probabilities?).

---

> ### Author Rebuttal · Authors · 2025-07-29
>
> We sincerely thank the reviewer for their positive comments and time in evaluating our paper. In the subsequent revisions, we shall make sure to make the presentation of our paper more appealing to the general audience. Below, we answer the question that you raised in the weaknesses section.
>
> > I think another thing is that for readers that haven't read the Fishelson et al. (2025) paper to define pseudo-calibration more formally (specifically $\text{PCal}$) and give a more gentle explanation for what intuition one should walk in with about why relaxing to pseudo-notions of calibration allows for the $\sqrt{T}$ - $T^{1/3}$ gap (does the noise in realizing the forecasting's predictions in non-pseudo calibration cause one to naively pay some variance term of order $\sqrt{T}$? Why does such a variance term exist, especially if the forecaster only needs to randomized over two very close probabilities?).}
>
> Great question. As you correctly figured out, the intuition behind the improved bound in $\text{PCal}_{2}$ is that by not taking into account the variance due to the randomness in the predictions, pseudo calibration is often easier to minimize.
>
> For the other part of the question, the forecasting algorithm of Fishelson et al. or our Algorithm 3 (both of which achieve $\text{Cal}_2 = \tilde{\mathcal{O}}(T^{\frac{1}{3}})$) do not randomize over only two adjacent probabilities. Both the algorithms are based on the BM-reduction, which requires computing the stationary distribution of a matrix $Q_t$, i.e., $p_t \in \Delta(K + 1)$ such that $Q_t p_t = p_t$ (subsequently, the forecaster outputs conditional distribution $\mathcal{P}_t$ with $\mathcal{P}_t(z_i) = p_t(i)$); the columns of $Q_t$ correspond to the probability distributions output by the $K + 1$ external regret algorithms invoked by the BM-reduction. While each external regret algorithm outputs a probability distribution that is supported on two adjacent probabilities in the discretization ${\mathcal{Z}}$, the stationary distribution $p_t$ need not necessarily be so. As shown in Fishelson et al. (or our Theorem 2), the resulting algorithm achieves $\text{PCal}_2 = \tilde{\mathcal{O}}(T^{\frac{1}{3}})$. Accounting for the variance in the predictions naively would incur a $\sqrt{T}$ variance term. However, as we show in Appendix E (Theorem 4), by applying Freedman's inequality this variance term is of the order $\tilde{\mathcal{O}}(|\mathcal{Z}|)$; particularly, we show that $\text{Cal}_2 \le 6\text{PCal}_2 + 96 |\mathcal{Z}| \log \frac{4|\mathcal{Z}|}{\delta}$ with probability at least $1 - \delta$. Substituting $|\mathcal{Z}| = \tilde{\Theta}(T^{\frac{1}{3}})$, we obtain an improved $\tilde{\mathcal{O}}(T^{\frac{1}{3}})$ variance bound and $\text{Cal}_2 = \tilde{\mathcal{O}}(T^{\frac{1}{3}})$.
>
> In summary, relaxing to pseudo calibration does enable achieving the $T^{\frac{1}{3}}$ rate, however, as we show this does not lead to a $\sqrt{T}-T^{\frac{1}{3}}$ ($\text{Cal}_2$ vs $\text{PCal}_2$) gap, since the variance term can be carefully controlled by a sophisticated analysis.
>
> **PS**: In light of you mentioning randomization over two very close probabilities, that is certainly true for some calibration algorithms, e.g., the forecaster by Foster (1999), or the forecaster in the expository note by Roth (2022) (ref. Section 3.4 in Chapter 3). However, the former results in $\text{Cal}_1 = \tilde{\mathcal{O}}(T^{\frac{2}{3}})$ and the latter implies $\text{Cal}_2 = \tilde{\mathcal{O}}(\sqrt{T})$, both of which do not imply the desired ${T}^{\frac{1}{3}}$ bound for $\text{Cal}_2$.
>
> Foster, D. P. (1999). A proof of calibration via Blackwell's approachability theorem. Games and Economic Behavior, 29(1-2), 73-78.
>
> Roth, A. (2022). Uncertain: Modern topics in uncertainty estimation. Unpublished Lecture Notes, 11, 30-31.
>
> We hope that our response answers your questions. Should you have any more questions, do not hesitate to reach out.

---

> > ### Comment · Reviewer_pP17 · 2025-08-06
> >
> > Thanks for the clarification! I continue to vote for acceptance

---

### Official Review · Reviewer_QAmB · 2025-07-16

**Clarity:** 3
**Significance:** 4
**Originality:** 4
**Rating:** 5
**Confidence:** 3

**Summary:**

The paper concerns the problem of online calibration. Its central contribution is the introduction and analysis of (pseudo) KL-Calibration, defined as (pseudo) calibration with respect to the logarithmic loss. The authors prove $\tilde{O}(T^{1/3})$ bounds on KL-Calibration and provide an explicit algorithm achieving the same rate for pseudo KL-Calibration. Subsequently, using a reduction scheme from KL divergence to other Bregman divergences, the authors extend these bounds to broader classes of losses, including bounded proper losses ($\mathcal{L}$), twice continuously differentiable losses ($\mathcal{L}_2$), and $G$-smooth losses ($\mathcal{L}_G$).

**Questions:**

What's the total computational complexity of the algorithm proposed in Section 5, based on the BM reduction?

Your algorithm requires knowledge of the time horizon $T$ to select an appropriate discretization level $K$. Is there a straightforward method (such as a doubling trick or similar approach) to avoid this requirement?

**Ethical Concerns:**

["NO or VERY MINOR ethics concerns only"]

**Final Justification:**

All my questions were positively resolved by the authors' feedback. That is why I am glad to keep my original score (accept).

**Limitations:**

yes

**Paper Formatting Concerns:**

no concerns

**Quality:**

3

**Strengths And Weaknesses:**

Strengths:
- A new concept of (pseudo) KL-Calibration, and proving its equivalence to minimizing (pseudo) swap regret with respect to the log loss.
- Bounding the (pseudo) swap regret relative to any twice continuously differentiable bounded proper loss by the (pseudo) KL-Calibration
- Similarly for any $G$-smooth loss, bounding the (pseudo) swap regret by the (pseudo) KL-Calibration with explicit constant $G$ in the bound.
- Proving the existence of an algorithm achieving $\tilde{O}(T^{1/3})$ KL-Calibration, along with providing an explicit algorithm achieving the same rate for pseudo KL-Calibration. The authors use an appropriate non-uniform discretization (resulting from the arcsin transform), ensuring that KL divergence values between neighboring discretization points are approximately uniform. This step seems crucial in controlling discretization error.
- High probability bound for maximum swap regret against smooth losses

Overall, I think the paper makes a significant contribution to the theory of online calibration. It generalizes previous results, particularly those of Fishelson et al. (2025), extending them to a broader class of loss functions, such as those induced by Tsallis entropy and the log loss (KL divergence). I think the main results, particularly the $\tilde{O}(T^{1/3})$ bounds on (pseudo) swap regrets, represent state-of-the-art advancement. Although many of the findings build upon previous results, the proofs contain sufficient novelty and introduce non-standard techniques that add considerable value to the work. In particular, the authors introduce several interesting tools, such as a non-uniform discretization (used in the past, but in a different context) and novel rounding schemes, which both seem crucial for effectively handling the log loss.

Weaknesses:
The only weakness that comes to my mind is that the paper is quite dense with mathematical content and technical detail, making it challenging to follow at times. It feels that without page-limit constraints, the presentation of these results could easily expand to twice the length.

I am a bit uncomfortable with the notation (e.g. in line 31 at page 1, but repeated several times afterwards) $\sum_{p \in [0,1]}$, which looks like a sum over uncountably many elements. Is it really a standard notation from past work?

---

> ### Author Rebuttal · Authors · 2025-07-30
>
> We sincerely thank the reviewer for their positive evaluation of the paper. Below, we provide responses to all your questions.
>
> > I am a bit uncomfortable with the notation (e.g. in line 31 at page 1, but repeated several times afterwards), which looks like a sum over uncountably many elements. Is it really a standard notation from past work?
>
> Yes, it is a standard notation from past work. The summation over $p \in [0, 1]$ is really just over the predictions $p_{1}, \dots, p_{T}$, since the interaction between the forecaster and the adversary lasts for $T$ rounds.
>
> > What's the total computational complexity of the algorithm proposed in Section 5, based on the BM reduction?
>
> The cost of the algorithm proposed in Section 5 at every time step is at most $\mathcal{O}(K ^ {2} + \text{INT} + \text{ST})$, where $\text{ST}$ is the time required to compute the stationary distribution of $Q_{t}$ (ref. Algorithm 3 in Appendix D) and $\text{INT}$ denotes the computation required for evaluating the integral $\frac{\int_{0} ^ {1} w \mu_{t, i}(w) dw}{\int_{0} ^ {1} \mu_{t, i}(w) dw}$ in line 3 of Algorithm 4 (Appendix D); the $\mathcal{O}(K^{2})$ cost is incurred in forming the matrix $Q_{t}$, and all other operations in Algorithm 3 can be carried out in time that is no worse than $\mathcal{O}(K^{2})$. For $\text{ST}$, the stationary distribution of $Q_{t}$ can be computed by the method of power iteration; notably, each iteration shall incur cost $\mathcal{O}(\text{nnz}(Q_{t}))$, where $\text{nnz}(Q_{t})$ represents the number of non-zero entries in $Q_{t}$. Since each column of $Q_{t}$ has at most two non-zero entries (Algorithm 2 randomizes over two adjacent points in the discretization), $\text{nnz}(Q_{t}) = \Theta(K)$. For $\text{INT}$, the integral is over $[0, 1]$ and has a closed-form expression in terms of the gamma function $\Gamma(z) \coloneqq \int_{0} ^ {1} \exp(-t) t^{z - 1} dt$ as derived below. Recall that \begin{align*}f_{\tau, i}(w) = p_{\tau, i} \ell(w, y_{\tau}) = -p_{\tau, i} (y_{\tau} \log w + (1 - y_{\tau}) \log (1 - w)) = \log (w ^ {-y_{\tau} p_{\tau, i}} (1 - w) ^ {-p_{\tau, i} (1 - y_{\tau})}).\end{align*}
> Therefore, $\mu_{t, i}(w) = \exp\left(-\sum_{\tau = 1} ^ {t - 1} f_{\tau, i}(w)\right) =  w^{\sum_{\tau = 1} ^ {t - 1} y_{\tau} p_{\tau, i}} (1 - w) ^ {\sum_{\tau = 1} ^ {t - 1} p_{\tau, i} (1 - y_{\tau})}$. For convenience, let $\gamma \coloneqq \sum_{\tau = 1} ^ {t - 1} y_{\tau} p_{\tau, i}, \delta \coloneqq \sum_{\tau = 1} ^ {t - 1} p_{\tau, i} (1 - y_{\tau})$. Then, \begin{align*}\int_{0} ^ {1} \mu_{t, i} (w) dw = \int_{0} ^ {1} w ^ {\gamma} (1 - w) ^ {\delta} dw = \text{B}(\gamma + 1, \delta + 1),\end{align*} where $\text{B}(z_{1}, z_{2})$ denotes the beta function, defined as $\text{B}(z_{1}, z_{2}) \coloneqq \int_{0} ^ {1} t ^ {z_{1} - 1} (1 - t) ^ {z_{2} - 1} dt$. Since $\text{B}(z_{1}, z_{2}) = \frac{\Gamma(z_{1}) \Gamma(z_{2})}{\Gamma(z_{1} + z_{2})}$ for all $z_{1}, z_{2} > 0$, we have $$\int_{0} ^ {1} \mu_{t, i}(w) dw = \frac{\Gamma(\gamma + 1) \Gamma(\delta + 1)}{\Gamma(\gamma + \delta + 2)}.$$ Similarly, $$\int_{0} ^ {1} w \mu_{t, i}(w) dw = \int_{0} ^ {1} w ^ {\gamma + 1} (1 - w) ^ {\delta} dw = \text{B}(\gamma + 2, \delta + 1) = \frac{\Gamma(\gamma + 2) \Gamma(\delta + 1)}{\Gamma (\gamma + \delta + 3)}.$$ Taking ratio of the two integrals above and using the identity $\Gamma(z + 1) = z \Gamma (z)$, which holds for all $z > 0$, we obtain \begin{align*}
>     \frac{\int_{0} ^ {1} w \mu_{t, i}(w) dw}{\int_{0} ^ {1} \mu_{t, i}(w) dw} = \frac{\Gamma (\gamma + 2)}{\Gamma (\gamma + 1)} \cdot \frac{\Gamma (\gamma + \delta + 2)}{\Gamma (\gamma + \delta + 3)} = \frac{\gamma + 1}{\gamma + \delta + 2} = \left(1 + \frac{\delta + 1}{\gamma + 1}\right) ^ {-1}.
> \end{align*}
> Clearly, at each time $t$, both $\gamma$ and $\delta$ can be computed in $\mathcal{O}(1)$ time using the previously memorized values corresponding to time $t - 1$. Therefore, $\text{INT} = \mathcal{O}(1)$. Since $K = \tilde{\Theta}(T^{\frac{1}{3}})$, the overall computation cost over $T$ rounds is $\tilde{\mathcal{O}}(T^{\frac{5}{3}} + T \cdot \text{ST})$.
>
> We thank the reviewer for the question and shall update our manuscript accordingly to incorporate the above discussion.
>
> > Your algorithm requires knowledge of the time horizon to select an appropriate discretization level. Is there a straightforward method (such as a doubling trick or similar approach) to avoid this requirement?
>
> Yes, the doubling trick can be utilized to avoid this requirement. Since KL-Calibration is equivalent to swap regret of the log loss, we can use the doubling trick to avoid the requirement of knowing the time horizon $T$. The analysis of doubling trick for swap regret is exactly identical to that for external regret.
>
> We hope that our responses answer your questions. If you have any further questions, please do not hesitate to reach out.

---

> > ### Comment · Reviewer_QAmB · 2025-08-05
> >
> > Dear Authors,
> >
> > Thank you for such a detailed response!
> > Actually, I would suggest including your derivations in the appendix of the final version of the paper, as they provide valuable clarification.
> >
> > I maintain my positive evaluation of your work.
> >
> > Best wishes,
> > Reviewer QAmB

---

### Comment · Area_Chair_e3dr · 2025-08-05
**Author-Reviewer Discussion Phase**

Dear Authors and Reviewers,

Thank you for supporting NeurIPS 2025!

We are now in the final two days of the discussion phase. Please make the most of this time to exchange your views!

I encourage each Reviewer to read the Authors' responses and provide a reply, if you have not already done so.

Thank you,

NeurIPS 2025 Area Chair

---

### Decision · Program_Chairs · 2025-09-17

**Decision:**

Accept (spotlight)

**Comment:**

The paper studies the problem of online calibration. The Authors introduce and analyze KL-calibration, defined with respect to the logarithmic loss. They prove the existence of an algorithm that achieves $\tilde O(T^{1/3})$ KL-calibration, and further provide an explicit algorithm attaining the same rate for the so-called pseudo KL-calibration, which is defined using the forecaster’s conditional distribution instead of the actual prediction in the calibration error. Finally, the results are generalized to a broader class of proper losses.

All Reviewers agree that this is a very solid theoretical paper presenting novel results. Only minor weaknesses were raised, such as the paper being dense and hard to follow, providing only the existence proof for KL-calibration, or the somewhat incremental nature of the results. Nevertheless, the Reviewers unanimously conclude that the results are worth publishing at NeurIPS.